# Melanoblast transcriptome analysis reveals pathways promoting melanoma metastasis

Kerrie L. Marie[1], Antonella Sassano[1,11], Howard H. Yang[1,11], Aleksandra M. Michalowski[1], Helen T. Michael[1], Theresa Guo[1,2], Yien Che Tsai[3], Allan M. Weissman[3], Maxwell P. Lee[1], Lisa M. Jenkins [4], M. Raza Zaidi [5], Eva Pérez-Guijarro[1], Chi-Ping Day[1], Kris Ylaya[6], Stephen M. Hewitt[6], Nimit L. Patel [7], Heinz Arnheiter[8], Sean Davis [9], Paul S. Meltzer[9], Glenn Merlino[1]* & Pravin J. Mishra[1,10]

Cutaneous malignant melanoma is an aggressive cancer of melanocytes with a strong propensity to metastasize. We posit that melanoma cells acquire metastatic capability by adopting an embryonic-like phenotype, and that a lineage approach would uncover metastatic melanoma biology. Using a genetically engineered mouse model to generate a rich melanoblast transcriptome dataset, we identify melanoblast-specific genes whose expression contribute to metastatic competence and derive a 43-gene signature that predicts patient survival. We identify a melanoblast gene, *KDELR3*, whose loss impairs experimental metastasis. In contrast, *KDELR1* deficiency enhances metastasis, providing the first example of different disease etiologies within the KDELR-family of retrograde transporters. We show that KDELR3 regulates the metastasis suppressor, KAI1, and report an interaction with the E3 ubiquitin-protein ligase gp78, a regulator of KAI1 degradation. Our work demonstrates that the melanoblast transcriptome can be mined to uncover targetable pathways for melanoma therapy.

[1] Laboratory of Cancer Biology and Genetics, Center for Cancer Research, National Cancer Institute, National Institutes of Health, Bethesda, MD 20892, USA. [2] Department of Otolaryngology—Head and Neck Surgery, Sidney Kimmel Comprehensive Cancer Center, Johns Hopkins Medical Institutions, Baltimore, MD 21287, USA. [3] Laboratory of Protein Dynamics and Signaling, Center for Cancer Research, National Cancer Institute, Frederick, MD 21702, USA. [4] Laboratory of Cell Biology, Center for Cancer Research, National Cancer Institute, National Institutes of Health, Bethesda, MD 20892, USA. [5] Fels Institute for Cancer Research and Molecular Biology, Lewis Katz School of Medicine at Temple University, Philadelphia, PA 19140, USA. [6] Experimental Pathology Laboratory, Center for Cancer Research, National Cancer Institute, National Institutes of Health, Bethesda, MD 20892, USA. [7] Small Animal Imaging Program, Frederick National Laboratory for Cancer Research, Leidos Biomedical Research Inc., Frederick, MD 21702, USA. [8] Mammalian Development Section, National Institute of Neurological Disorders and Stroke, National Institute of Health, Bethesda, MD 20892, USA. [9] Genetics Branch, Center for Cancer Research, National Cancer Institute, National Institutes of Health, Bethesda, MD 20892, USA. [10] James Cancer Hospital and Solove Research Institute, Ohio State University Comprehensive Cancer Center, Columbus, OH 43210, USA. [11] These authors contributed equally: Antonella Sassano, Howard H. Yang.
*email: gmerlino@helix.nih.gov

Melanoma is an aggressive cancer that frequently progresses to metastatic proficiency. Treatment of metastatic melanoma remains a challenge, highlighting an urgent need to uncover new targets that could be used in the clinic to broaden therapeutic options. In the early nineteenth century, Virchow[1] first described cancer cells as being embryonic-like. Developmental systems have since proven useful to study melanoma, and melanoma cell plasticity is a key feature of melanoma progression. Melanocyte lineage pathways are a recurring theme in melanoma etiology, reinforcing the importance of uncovering new melanocyte developmental biology[2–11]. Here we use a genetically engineered mouse (GEM), designed to facilitate the isolation and analysis of developing melanocytes (melanoblasts), to attempt to uncover targets relevant to melanoma metastasis.

Melanocytes are neural crest-derived cells whose development necessitates extensive migration/invasion to populate the skin and other sites[12]. This process requires melanoblasts to adopt a migratory phenotype, to interact with and survive in foreign microenvironments, and to colonize distant sites—functions that are analogous to metastatic competence[13]. To complete these processes, the cell may encounter numerous cellular stressors, such as shear stress, nutrient deprivation, hypoxia, lipid stress, and oxidative stress[14]. The cellular impact of these stressors converges at the endoplasmic reticulum (ER), an organelle tied closely to protein synthesis and responsible for correct folding, quality control, and the post-translational modification of the cellular proteins that enter the secretory pathway. Stress stimuli results in aberrant ER function, a buildup of unfolded/misfolded proteins (ER stress), and an overwhelmed system. The ER can therefore be viewed as an exquisitely sensitive stress sensor. Upon ER stress insult, the ER launches an immediate counter measure known as the unfolded protein response (UPR)[15]. The UPR consists of three arms, the IRE1, PERK, and ATF-6 pathways. Cumulatively, these result in transcriptional activation of chaperones and ER-associated degradation (ERAD) machinery that target unfolded proteins for degradation to help counter the stress[15]. Simultaneously, the PERK pathway attenuates translation to reduce protein load in the ER. Unchecked ER stress can result in cell death via the PERK-stimulated CHOP (CCAAT-enhancer-binding protein homologous protein) pathway[15]. The KDEL receptors (KDELRs) are a family of seven-transmembrane-domain ER protein retention receptors consisting of three members (KDELR1, 2, and 3) that function in the ER stress response (ERSR). They share structural homology, but each isoform can have different ligands[16,17]. They are responsible for the retrograde transport of protein machinery from the Golgi to the ER, including chaperones that target unfolded proteins for refolding, and whose disassociation from membrane receptors stimulates UPR signaling[17,18]. In embryogenesis, there is a need for tightly coordinated temporal control of gene/protein expression for correct differentiation of tissues[14]. Embryonic cells are therefore primed to accommodate overwhelming ER stress, as this would affect the cell's ability to translate, synthesize, fold, and modify proteins, which would otherwise compromise the developing embryo[14].

We hypothesize that genes whose expression is upregulated in developing melanoblasts and metastatic melanoma, but downregulated in differentiated melanocytes (hereafter referred to as MetDev genes), can be reactivated by melanoma cells to facilitate metastasis (Fig. 1a). To explore this, we use a GEM model in which green fluorescent protein (GFP) is inducibly targeted to embryonic melanoblasts and mature melanocytes through the dopachrome tautomerase (*Dct*) promoter to drive expression (inducible *Dct*-GFP; i*Dct*-GFP)[19]. This powerful tool enables identification/isolation of cells of the melanocytic lineage[19], useful

for investigation of the melanoblast transcriptome. Employing this approach, we identify a 43-gene embryonic melanoblast signature that predicts metastatic melanoma patient survival, and we introduce a role for *KDELR3* that is distinct from *KDELR1*. A metastasis suppressor screen highlights KAI1/CD82 (hereafter referred to as KAI1) as a KDELR3-regulated protein. We observe that KDELR3 regulates KAI1 protein levels and post-translational modification. We discover an interaction between KDELR3 and gp78, the E3 ubiquitin-protein ligase known to regulate KAI1 degradation[20]. Our work shows that melanoma cells can commandeer embryonic transcriptomic programs to promote their progression to metastasis. These genes represent an untapped source of targetable pathways to exploit for improving melanoma treatment.

## Results

**Melanoblast transcriptomic expression in melanoma metastasis.** To study melanoblast genes, GFP-positive melanocytic cells were isolated from four developmental time points: embryonic days (E) 15.5 and 17.5 and postnatal days (P) 1 and 7 (Fig. 1b, Supplementary Fig. 1a, b). These four stages represent embryonic melanoblast development from the neural crest into differentiated quiescent melanocytes of the postnatal pup[21,22]. Melanocytes/melanoblasts were isolated by using fluorescence-activated cell sorting (FACS) from i*Dct*-GFP mice (Supplementary Fig. 1c). At E15.5 and E17.5, melanoblasts are migrating and colonizing the hair follicles within the epidermis[21,23]—processes that are highly relevant to metastasis, particularly to colonization at the metastatic site—and intrafollicular melanoblasts are still present[23]. P1 and P7 mature melanocytes were selected as a model of differentiated melanocytes. Melanocytic cells were extracted from multiple litters (6–10 pups) at each developmental stage to ensure comprehensive representation of all melanoblasts/melanocytes present. RNA was extracted for whole-transcriptome sequencing.

Genes with differential expression between embryonic melanoblasts (E15.5 and E17.5) and postnatal differentiated melanocytes (P1 and P7) were identified by using DE-seq2[24] with a $q$ value <0.1, and filtered for genes with $\log_2$ fold change >1.5, indicating an increase in gene expression in melanoblasts over melanocytes. We reasoned that a fold change less than this was less likely to be biologically meaningful. Four-hundred and sixty-seven melanoblast-specific genes were identified from our analyses, which we hypothesize to be putative melanoma metastasis enhancer genes (MetDev genes; Fig. 1c; Supplementary Fig. 2a). If our hypothesis is correct, we should be able to identify melanoblast-specific genes that are upregulated in metastases compared with primary tumors. Our analyses confirmed that 76 MetDev genes were upregulated in stage III/IV metastatic melanoma samples compared with stage I/II primary tumor samples (Supplementary Fig. 3a; GSE8401)[25]. These 76 genes were then validated in a secondary patient dataset, which showed that increased MetDev gene expression correlated significantly with more advanced melanoma stage (Supplementary Fig. 3b; GSE98394)[26]. While analysis of differential expression across treatment-naive patient samples is informative of metastatic biology, we wanted to address specifically how our MetDev genes contribute to patient progression in the clinic. To this end, we interrogated our 467 putative MetDev genes by using a Cox proportional hazards model to associate their expression with overall survival in a training dataset of human patient samples derived from melanoma metastases (stages III and IV; GSE19234)[27]. We discerned a 43-gene survival risk predictor (Fig. 1c, d) that could accurately predict patient outcome in a separate testing dataset of late-stage (stages III and IV) metastatic melanoma patient samples derived from metastases (GSE8401; Fig. 1e)[25]. These data show that our MetDev cohort is enriched for

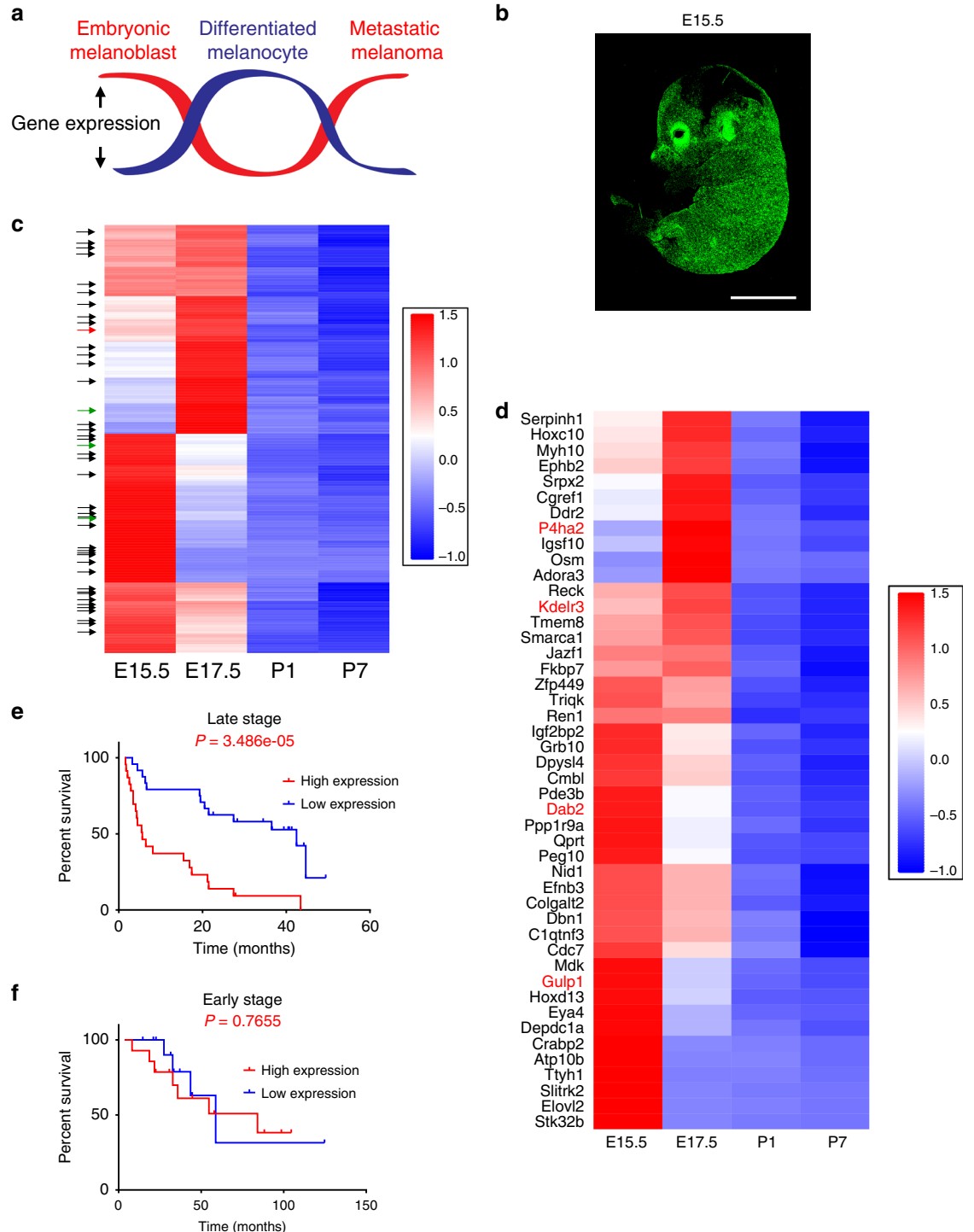

**Fig. 1 Discovery of metastasis development (MetDev) genes. a** Schematic depicting the experimental hypothesis: genes whose expression is upregulated in melanoblasts and metastatic melanoma, but downregulated in differentiated melanocytes (red line), may drive cellular functions that promote melanoma metastasis (MetDev genes). **b** Confocal imaging of *iDct-GFP* embryo at embryonic day 15.5 (E15.5) is magnification ×5, scale bars, 5 mm. **c** RNA-seq expression of mouse developing melanocytes: 467 embryo-specific genes shown. Black arrows: 42 genes identified from Cox proportional hazards model. Green arrows: genes functionally validated. Red arrow: *Kdelr3*. *Kdelr3* validated both in Cox proportional hazards model and functionally validated. Embryonic days 15.5 and 17.5 (E15.5 and E17.5, respectively). Postnatal day 1 and postnatal day 7 (P1, P7, respectively). **d** RNA-seq expression of 46 genes in mouse developing melanocytes: Black text: 42 genes identified from Cox proportional hazards model. Red text: four genes functionally validated. *Kdelr3* validated both in Cox proportional hazards model and functionally validated. **e, f** Cox proportional hazards modeling (GSE19234) yielded a 43-gene MetDev signature. Patients' risk assessed in GSE8401 patient cohort. Late stage: stage III/IV metastatic melanomas. Early stage: stage I/II primary tumors. High expression: high expression of gene signature. Low expression: low expression of gene signature. Log-rank test. Late stage, high ($N = 23$) vs. low ($N = 24$), $P = 3.486\mathrm{e} - 05$. Early stage, high ($N = 14$) vs. low ($N = 13$), $P = 0.7655$. **c, d** Color scales represent gene expression z scores.

**Table 1 siRNA screen for metastatic potential of four putative MetDev genes.**

| Gene symbol | Colony formation | Tail vein metastasis |
|---|---|---|
| Gulp1 | P = 0.0002 | P = 0.0002 |
| Kdelr3 | P = 0.0122 | P = 0.0155 |
| P4ha2 | P = 0.022 | P = 0.022 |
| Dab2 | P = 0.0428 | P = 0.0421 |

siRNA knockdown of genes indicated (B16 cell line). Colony formation assay, n = 10 wells (Dab2, Kdelr3, control), n = 5 wells (Gulp1, P4ha2, control), screen performed once. P value assessed by Kruskal–Wallis using uncorrected Dunn's test vs. siControl. Tail vein metastasis assay, n = 10 mice (Dab2, Kdelr3, control), n = 5 mice (Gulp1, P4ha2, control), screen performed once. P value assessed by Kruskal–Wallis using uncorrected Dunn's test vs. siControl

metastatic progression genes and can also predict survival in multiple independent patient datasets. Notably, gene expression levels in samples derived from early-stage (stages I and II) primary melanoma lesions did not predict patient outcome, suggesting that MetDev genes play a key role in late-stage disease specifically (GSE8401; Fig. 1f)[25].

To allow functional validation of our MetDev candidates in both soft agar colony-forming assays and in experimental metastasis models, we prioritized the list of MetDev gene candidates. To do this, we applied criteria based solely on melanoblast expression data, selecting for genes with no detectable gene expression in P7 postnatal pups. Differential expression was validated using a microarray expression dataset derived from our iDct-GFP model (E17.5 vs. P2 and P7; q value <0.1, linear regression model)[19]. Further criteria using differences in fold-increase expression in melanoblasts vs. melanocytes and the greatest expression at embryonic stages allowed us to select 20 genes most likely to be functionally relevant. Of these 20, we noted that seven genes (Kdelr3, P4ha2, Gulp1, Dab2, Lum, Aspn, and Mfap5) were associated with extracellular matrix (ECM) or trafficking, which we decided to focus on. For functional analyses, we chose three of these seven genes (Kdelr3, Gulp1, and Dab2; Fig. 1c, d), which we had shown were correlated with advanced disease (Supplementary Fig. 3a, b) and had no established role in cutaneous melanoma metastasis. As a positive control, we included P4ha2 (Fig. 1c, d), which is prognostic of worse clinical outcomes in melanoma and associated with metastasis in other cancers[28]. Small-interfering RNA (siRNA) knockdown of all four candidate genes in B16 mouse melanoma cells inhibited both growth in soft agar colony formation assays and formation of lung metastases in experimental metastasis assays compared with non-targeting controls (Table 1). Moreover, protein expression in human tumor microarrays (TMAs; the NCI melanoma progression microarray[29]; Supplementary Fig. 3c–h) confirmed KDELR3, P4HA2, and DAB2 expression all markedly increased with advancement of disease. Our work demonstrates that the MetDev dataset is enriched in genes that have a functional role in melanoma metastasis. We identify melanoma metastasis genes and highlight ECM and trafficking as important pathways common to both melanoblast development and melanoma metastasis.

We further observed significant co-expression of three of the four functionally validated genes (Kdelr3, P4ha2, and Dab2) throughout four distinct mouse models of melanoma (see the "Methods" section and Supplementary Table 1), corroborated in a melanoma patient cohort (The Cancer Genome Atlas (TCGA); Supplementary Table 2). Notably, expression of Kdelr3 and P4ha2 was highly correlated throughout all datasets (Supplementary Fig. 4a, b), raising the possibility that some MetDev genes may be co-regulated and serve a more coordinated role in metastasis.

**KDELR3 encodes a Golgi-resident protein whose expression correlates with melanoblast development and melanoma progression**. To understand how melanoblast genes might facilitate metastasis, we chose to study one MetDev gene in depth. KDELR3 was selected as it was a positive hit in all our analyses: KDELR3 encodes a trafficking protein important in the ERSR whose expression was associated with poor patient prognosis in metastatic melanomas (Fig. 1e, 43-gene signature), whose expression is upregulated during melanoma progression (Supplementary Fig. 3a–c, f) and was functionally validated in soft agar colony formation and experimental metastasis assays (Table 1). The KDELRs are Golgi-to-ER retrograde transporters responsible for maintaining ER localization of their protein substrates, which consist of protein chaperones required for protein folding and targeting unfolded proteins for degradation[18], thereby maintaining ER quality. We showed that KDELR3 is localized to both the cis- and trans-Golgi compartments in metastatic melanoma cells (Supplementary Fig. 4c) and validated expression of KDELR3 in mouse melanoblasts (Fig. 2a). Moreover, within the KDELR family only KDELR3 demonstrated a melanoblast-specific expression pattern and showed consistent upregulation in melanoma cell lines (Fig. 2b; Supplementary Fig. 4d, e). These data raise the possibility that KDELR3 plays a role in melanoma progression that is distinct from other KDELRs, despite their presumed redundancy. Analysis of human patient datasets and tumor histology microarrays confirmed an upregulation of KDELR3 expression in malignant melanoma vs. benign nevi (Fig. 2c–e).

We sought to functionally validate a role of KDELR3 in melanoma progression. We used human and mouse melanoma cells to demonstrate that siRNA and short-hairpin RNA (shRNA) knockdown of KDELR3 significantly reduced, and KDELR3 overexpression enhanced, anchorage-independent growth (Fig. 3a–d; Supplementary Fig. 5a, b), which cannot be attributed to a change in proliferation (Supplementary Fig. 5c). There are two KDELR3 variants, and we selected the KDELR3-001 variant to perform rescue experiments as it is the most abundant transcript expressed in human cell lines and patient samples. We therefore performed rescue experiments via exogenous expression of KDELR3-001$^{Mu}$, whose shRNA recognition site had been mutated without altering the final protein sequence. KDELR3-001$^{Mu}$ expression was restored, rescuing the anchorage-independent growth phenotype (Fig. 3e–g; Supplementary Fig. 5d). KDELR3 was therefore validated as a mediator of anchorage-independent growth in melanoma cells, a process required for metastasis.

**KDELR3 knockdown reduces lung colonization in experimental metastasis assays**. To assess the relevance of KDELR3 within the metastatic cascade, we used a tail vein experimental metastasis assay, which specifically assesses the ability of the cells to extravasate and colonize the lung, processes that are critical for metastasis and that may mimic normal hair follicle colonization (E17.5). Transient knockdown of KDELR3 in either mouse (Fig. 3h, i) or human melanoma cell lines (Fig. 3j, Supplementary Fig. 6a) resulted in significantly reduced metastatic potential compared with non-targeting controls, indicating that KDELR3 expression is important for the cells' ability to extravasate/colonize the lung, further validating that KDELR3 is a melanoblast gene that functions in metastasis (MetDev gene). Stable shRNA knockdown of KDELR3 also resulted in a reduction in lung colonization following tail vein metastasis and significantly fewer mice characterized with high metastatic burden (Supplementary Fig. 6b–f). However, no appreciable difference in cell cycle or subcutaneous in vivo tumor growth was observed

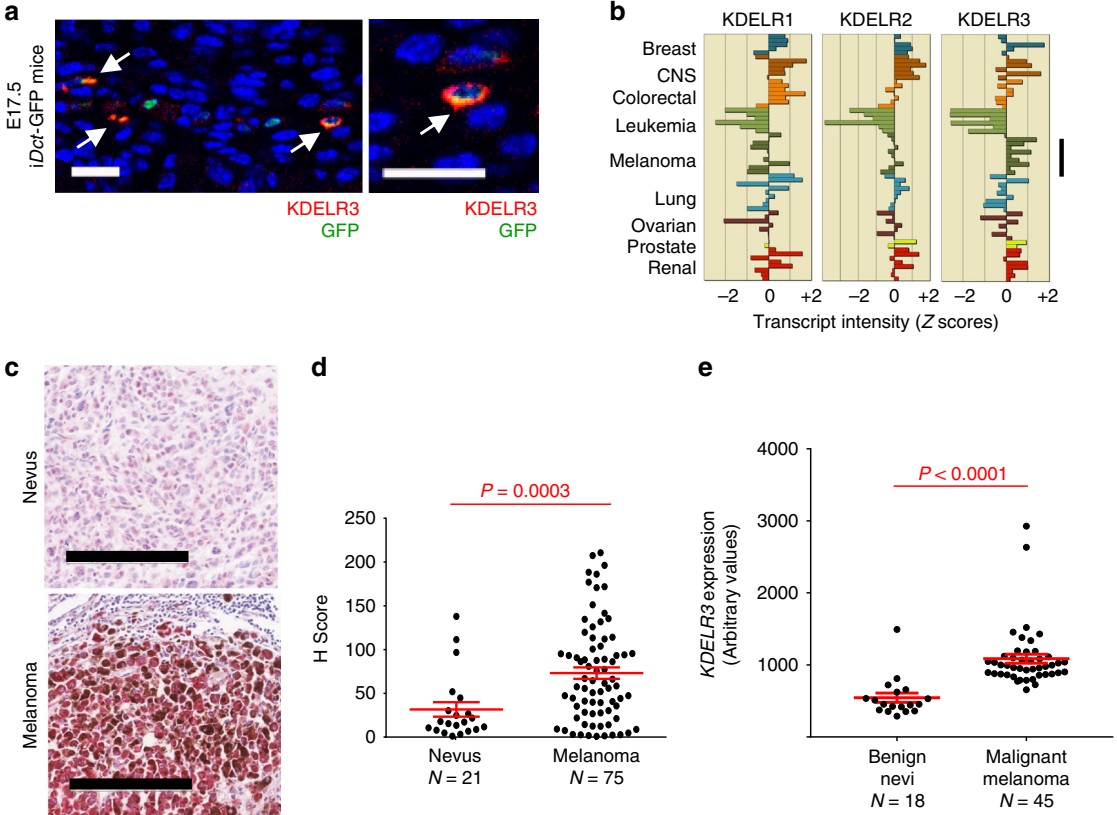

**Fig. 2 Melanoblast gene expression in melanoma. a** KDELR3 (red) and GFP (green) staining in E17.5 i*Dct*-GFP mouse skin. White arrows depict colocalization. Magnification, ×40. Scale bars, 20 μm. Representative image of 100 cells analyzed taken from one mouse. **b** Pan-cancer RNA expression of KDEL receptors in human cell lines (NCI60, CellMiner analysis); *KDELR3* expression in melanoma (black line). **c** *KDELR3* expression in human nevus and melanoma lymph node metastasis (red intercellular staining), magnification, ×10. Scale bar represents 200 μm. **d** H score of KDELR3 immunohistochemistry in human tumor microarrays. Unpaired two-tailed Student's *t* test with Welch's correction, $P = 0.0003$, d.f. $= 47.9$, $t = 3.936$. $N = 21$ (nevus) and $N = 75$ (melanoma). **e** *KDELR3* expression in benign nevi and malignant melanoma (GSE3189; 204017_at probeset). Unpaired two-tailed Student's *t* test with Welch's correction, $P < 0.0001$, d.f. $= 47.39$, $t = 6.035$. $N = 18$ (benign nevi) and $N = 45$ (malignant melanoma). **d**, **e** Line and error bars represent mean ± s. e.m.

(Supplementary Fig. 6g–i), suggesting that the *KDELR3*-mediated metastatic phenotype cannot be attributed to a change in proliferation, and that *KDELR3* is a genuine melanoma metastasis progression gene.

**KDELR3 and the ERSR in metastatic melanoma.** To uncover the role of *KDELR3* in melanoma metastasis, we asked which pathways were co-regulated with *KDELR3* expression. Gene set enrichment analysis (GSEA, false discovery rate (FDR) <0.0001) of *KDELR3*-co-expressed genes in TCGA skin cutaneous melanoma patients (cBioPortal)[30,31] revealed gene ontology (GO) term enrichment of ECM and trafficking pathways (consistent with previous data, Supplementary Figs. 2a and 7a), and pathways involved in the ERSR and response to unfolded proteins (Supplementary Fig. 7a). Quantitative mass spectrometry was used to analyze whole-cell lysates of *KDELR3* knockdown compared with non-targeting controls and parental controls; GSEA analysis revealed the top-scoring, most consistent pathway using GO term enrichment showed upregulation of ER lumen proteins (Supplementary Fig. 7b). Enriched proteins included protein chaperones, lectins, and enzymes involved in protein folding and targeting misfolded proteins for degradation (including UGGT, ER lectin, FKBP7, and calumenin), which is consistent with an increase in misfolded protein load in *KDELR3*-knockdown cells[32]. We therefore asked how *KDELR3*'s role in the ERSR response is

associated with its metastasis phenotype. Metastasis induces ER stress, UPR activation, and downstream signaling events function to alleviate this stress[15]. High doses of ER stress, or an ineffective UPR, have been associated with deleterious signals and ultimately cell death. We therefore hypothesized that one role of *KDELR3* in metastasis would be to alleviate ER stress-induced deleterious signaling (Supplementary Fig. 7c). We observed in four independent mouse models of melanoma ($N = 6$–13 mice per model) that *Perk* (*Eif2ak3*) transcription was negatively correlated with *Kdelr3* transcription (Fig. 4a), whereas *Gadd34* (*Ppp1r15a*) transcription was positively correlated (Fig. 4b). As PERK is a protein kinase and GADD34 a protein phosphatase, which both act on EIF2α[33], we hypothesized that KDELR3-low cells are primed to activate the PERK–EIF2α arm of the UPR. We knocked down *KDELR3* (KD) in both 1205Lu and WM-46 human cell lines (shRNA knockdown, Supplementary Fig. 6b) and found that loss of *KDELR3* expression resulted in increased PERK and EIF2α protein levels in untreated cells, corroborating our mouse model data (Fig. 4c). We also saw a concomitant increase in PERK and EIF2α phosphorylation, suggesting constitutive activation of the PERK–EIF2α axis in untreated KD cells (Fig. 4c). The other two branches of the UPR pathways, the IRE1–XBP1 and ATF-6α axes, were inactive in untreated *KDELR3* KD cells (Supplementary Fig. 7d, e). Tunicamycin, a chemical inhibitor of N-glycosylation that induces ER stress in cells, was used as a positive control (Fig. 4c; Supplementary Fig. 7d, e).

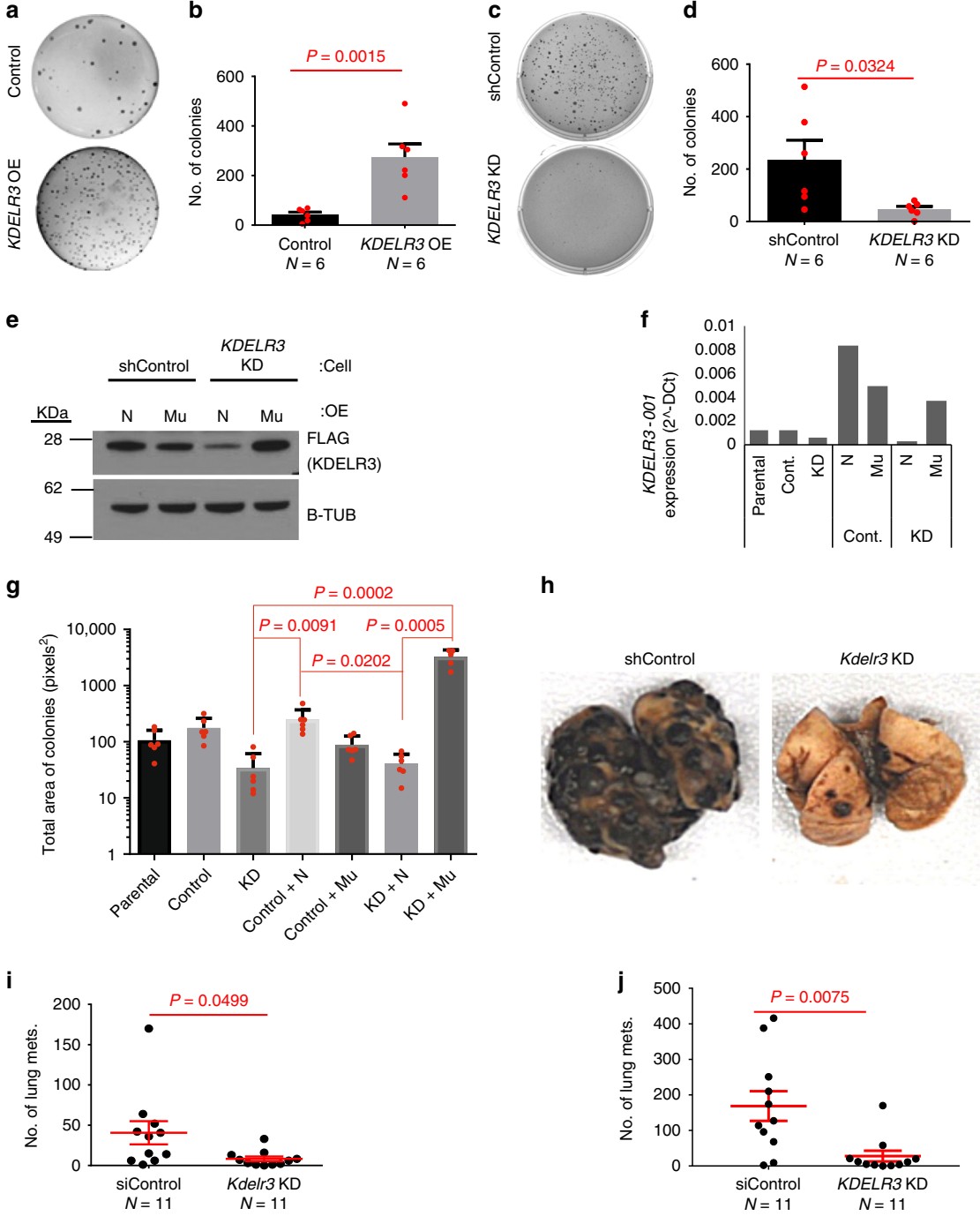

**Fig. 3 *KDELR3* mediates melanoma metastatic potential. a–d** Soft agar colony formation assay with **a**, **b** overexpression of *KDELR3* (*KDELR3* OE) in human SK-MEL-28 cells vs. parental cell, unpaired two-tailed Student's *t* test, *P* = 0.0015, d.f. = 10, *t* = 4.307. Six wells were analyzed per group. *N* = 6 (Control) and *N* = 6 (KDELR3 OE). **c**, **d** shRNA *KDELR3* knockdown (*KDELR3* KD) in human WM-46 cells vs. non-targeting control, unpaired two-tailed Student's *t* test, *P* = 0.0324, d.f. = 10, *t* = 2.483. *N* = 6 wells analyzed per group. **e**, **f** Western blot and qPCR analysis of exogenously expressed FLAG-tagged *KDELR3-001*; ENST00000216014 (N) and *KDELR3-001^Mu* (Mu) in WM-46 (**e**) and 1205Lu (**f**) cells, transduced with non-targeting control (shControl/Cont./Control) or *KDELR3*-targeted (KD) shRNAs. Total *KDELR3-001* RNA (*KDELR3-001* and *KDELR3-001^Mu*) (**f**). **g** Rescue of soft agar colony formation in *KDELR3-001^Mu* cells (WM-46), Kruskal–Wallis with Dunn's multiple comparison test. Five to six wells were analyzed per group. **h**, **i** Tail vein metastasis of *Kdelr3* siRNA knockdown (*Kdelr3* KD) in mouse B16 cells. Unpaired two-tailed Student's *t* test with Welch's correction, *P* = 0.0499, d.f. = 10.83, *t* = 2.207. *N* = 11 (siControl) and *N* = 11 (*Kdelr3* KD). **j** Tail vein metastasis of *KDELR3* siRNA-mediated knockdown in human 1205Lu cells transduced with *Ferh-luc-GFP*. Unpaired two-tailed Student's *t* test with Welch's correction, *P* = 0.0075, d.f. = 12.57, *t* = 3. *N* = 11 (siControl) and *N* = 11 (*KDELR3* KD). **b**, **d** Bars and error bars depict mean ± s.e.m. **g** Bars and error bars depict mean ± SD. **i**, **j** Lines and error bars depict mean ± s.e.m. **a–f**, **h–j** Representative of three independent experiments. **g** Representative of two independent experiments. **e** β-tubulin loading control.

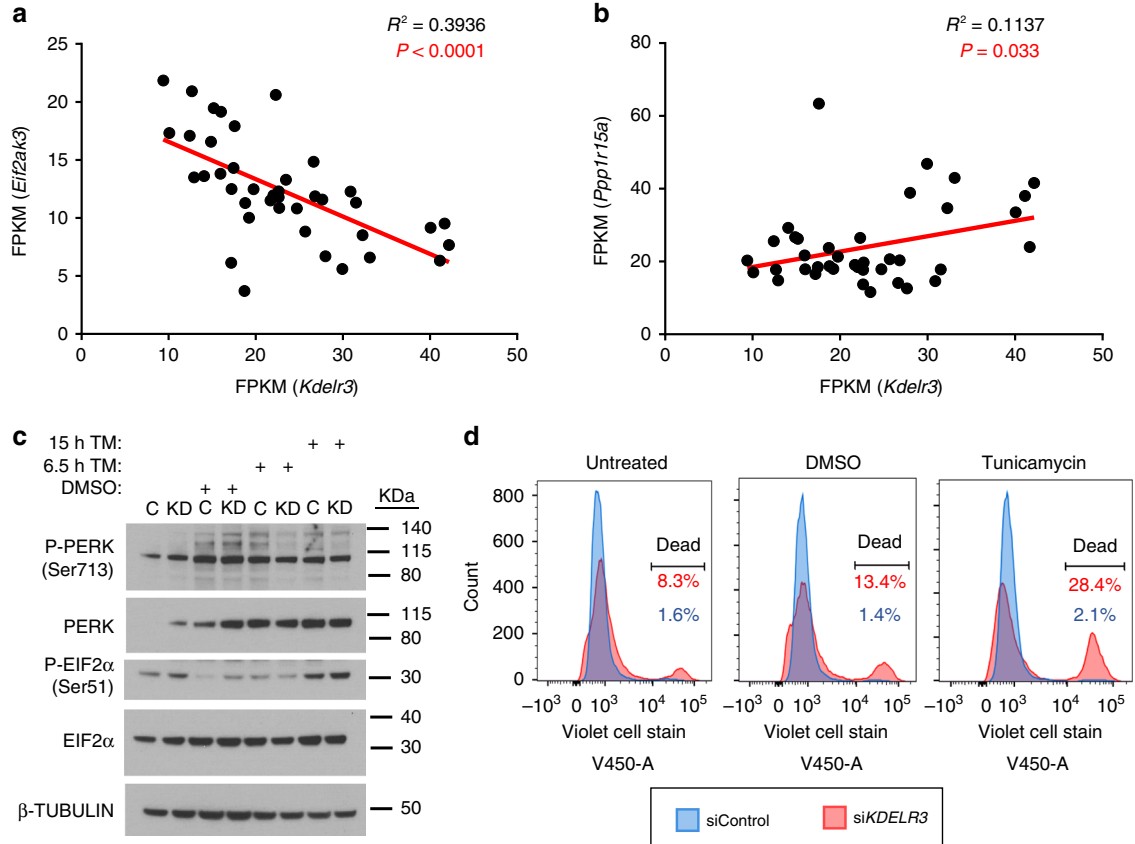

**Fig. 4 KDELR3 and the ER stress response in metastatic melanoma. a** Scatter plot of *Eif2ak3* (*Perk*) RNA expression vs. *Kdelr3* RNA expression. Linear regression analysis, $R^2 = 0.3936$, $P < 0.0001$. **b** Scatter plot of *Ppp1r15a* (*Gadd34*) RNA expression vs. *Kdelr3* RNA expression. Linear regression analysis, $R^2 = 0.1137$, $P = 0.03$. **a**, **b** Data from four independent mouse models of melanoma (see Methods). Each dot represents one mouse. M1, $N = 9$ mice; M2, $N = 6$ mice; M3, $N = 12$ mice; M4, $N = 13$ mice. **c** Western blot analysis of PERK and eIF2α signaling (1205Lu cells). Non-targeting control (shControl, C) and *KDELR3* knockdown (sh*KDELR3*, KD) in cells were untreated or treated with 3 µg/ml tunicamycin (TM) or DMSO control for the indicated time. Immunoblot with antibodies specified, β-tubulin loading control. **d** Live/dead violet cell stain in *KDELR3*-knockdown 1205Lu cells. Untreated, DMSO, and tunicamycin (2.5 µg/ml) treatment groups were treated 18 h before collection. The right-hand peak on the graph indicates the percentage of dead cells. Representative of three independent experiments (**c**, **d**).

Untreated *KDELR3* KD cells exhibited reduced levels of BiP, an essential protein chaperone necessary for activation of all arms of the UPR[15], suggesting that retrograde transport in non-stressed cell may be required for long-term maintenance of BiP homeostasis (Supplementary Fig. 7e)[17]. These data indicate that loss of *KDELR3* expression disrupted ER homeostasis, resulting in a dysregulated UPR, which has previously been linked with ER stress-associated cell death[34]. We hypothesized that *KDELR3* functions to alleviate deleterious ER stress-induced signaling (Supplementary Fig. 7c). To test this, we asked if *KDELR3* knockdown sensitizes metastatic melanoma cells to ER stress-induced death. We treated cells with tunicamycin, and measured cell death through flow cytometry using Live/Dead cell stain. We observed that siRNA-mediated knockdown of *KDELR3* expression resulted in a ~5-fold increase in metastatic melanoma cell death over controls (8.3%, si*KDELR3*; 1.6%, siControl; Fig. 4d). These data suggest that *KDELR3* promotes cell survival in metastatic melanoma cells, which likely influences metastatic potential. However, *KDELR3*-knockdown cells have an enhanced sensitivity to ER stress induction with tunicamycin (>13-fold difference in cell death: 28.4%, si*KDELR3*; 2.1%, siControl; Fig. 4d). If our hypothesis is correct, we expect *KDELR3* to be critical to metastatic melanoma viability, but not to normal melanocytes. To answer this, we used a primary melanocyte cell line (234), which was immortalized using hTERT expression and

p16 shRNA knockdown (234 hTERT-sh_p16)[35]. We found that, contrary to metastatic melanoma, loss of *KDELR3* in these cells is not critical for cell viability (Supplementary Fig. 8a, b). These data indicate that the ability of *KDELR3* to relieve ER stress is crucial for adaptation and survival of metastatic melanoma and may be instrumental to the metastatic phenotype.

**KDELR3 mediates post-translational regulation of the metastasis suppressor KAI1.** To further understand the role of *KDELR3* in metastasis, we queried if *KDELR3* knockdown would increase expression of known metastasis suppressors in melanoma. To address this, we screened protein expression of five melanoma metastasis suppressors (BRMS1, gelsolin, GAS1, NME1/NM23-H1, and KAI1) following *KDELR3* knockdown[36,37]. Of these, only KAI1 demonstrated a marked increase in expression following *KDELR3* knockdown (Fig. 5a). Moreover, we observed a change in KAI1 molecular weight distribution following *KDELR3* knockdown, suggesting alterations in KAI1 post-translational modification. KAI1 protein upregulation was independent of transcriptional changes (Fig. 5b), supporting a regulatory role for KDELR3 at the post-translational level. KAI1 has been shown to influence metastasis through multiple mechanisms, including cell–cell adhesion, cell motility, cell death, and senescence, and protein trafficking in many cancer types,

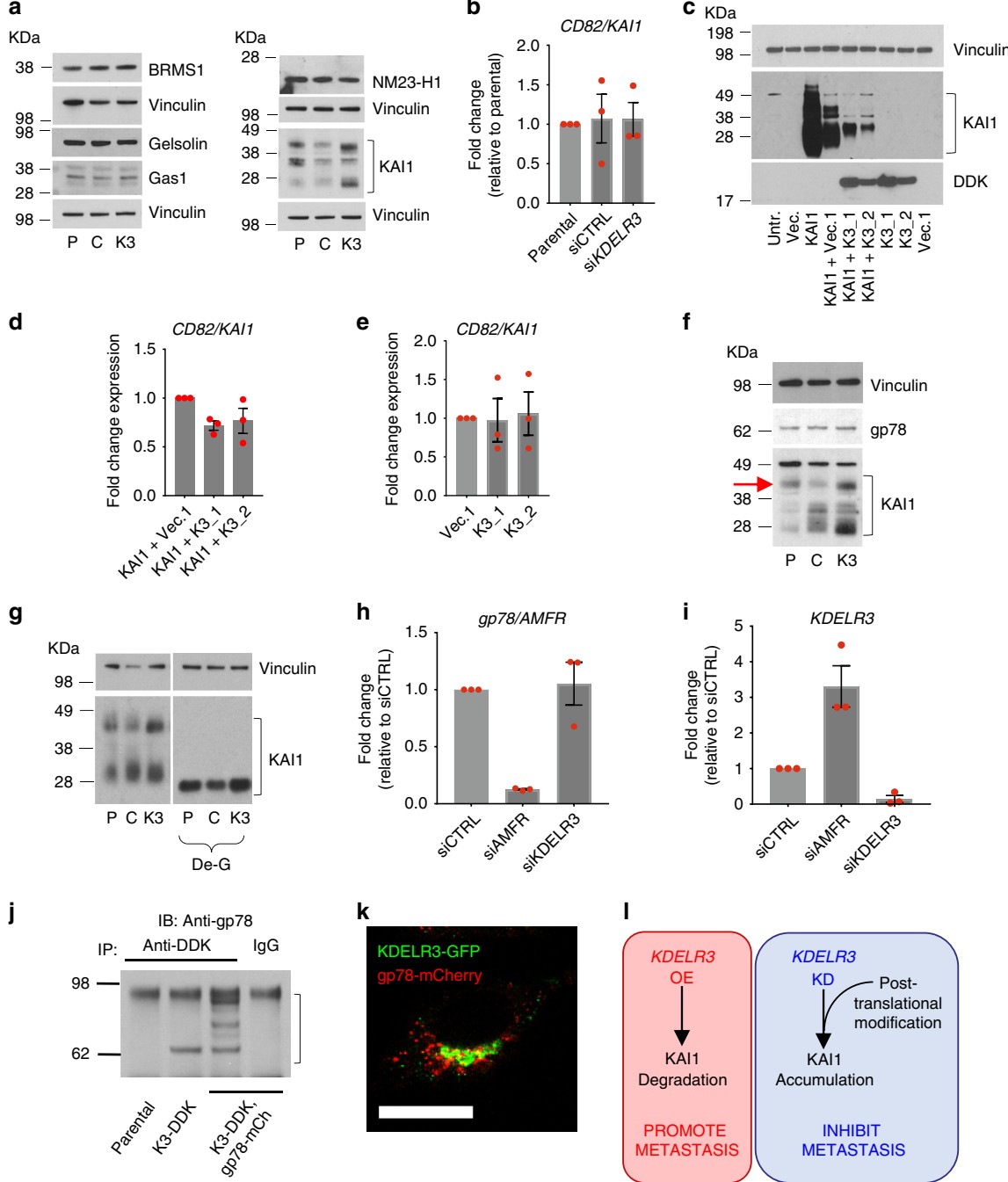

**Fig. 5 KDELR3 regulates expression and processing of the metastasis suppressor KAI1. a** Screen of known melanoma metastasis suppressor expression following *KDELR* knockdown (3 days post knockdown). P, parental; C, siControl; K3, si*KDELR3*. **b** qPCR of *KAI1* RNA expression (*CD82* gene) in siRNA-knockdown cells (indicated), 3 days post knockdown. **c–e** KAI1 protein (**c**) and RNA (**d**, **e**) expression in 1205Lu cells transfected with CD82/KAI1 overexpression (KAI1) or PCMV6-AC control vector (Vec.), *KDELR3* transcript 1 with DDK tag (K3_1), *KDELR3* transcript 2 with DDK tag (K3_2), or PCMV6 control vector (Vec.1). They were harvested 3 days post transfection. Equal protein amounts subjected to immunoblot analysis with an anti-KAI1 and anti-DDK antibody and anti-vinculin loading control (**c**). **f** 1205Lu cells parental (P), and 1205Lu cells transiently transfected with control siRNA (C), and *KDELR3* siRNA (K3), harvested 3 days post transfection and equal protein amounts subjected to immunoblot analysis with an anti-KAI1 and anti-gp78 antibody. Red arrow indicates high-molecular-weight KAI1. **g** KAI1 protein expression in siRNA-knockdown (indicated) 1205Lu cells harvested 3 days post transfection and treated with deglycosylation enzymes (De-G). **h** qPCR of gp78 RNA expression (*AMFR* gene) in siRNA-knockdown cells (indicated), 4 days post knockdown. **f**, **g** Anti-vinculin antibody used to control for protein loading. **i** qPCR of *KDELR3* RNA expression in siRNA-knockdown cells (indicated), 4 days post knockdown. **j** Co-immunoprecipitation of endogenous gp78 and mCherry-tagged gp78 (gp78-mCh) with FLAG-tagged KDELR3 (K3-DDK) in stably transduced 1205Lu cells. **k** *pol2 > KDELR3*-GFP (green) co-localizes with *pol2 > gp78*-mCherry (red) in 1205Lu metastatic melanoma cells. Scale bars, 50 μm. **l** Schematic of the KDELR3–KAI1 axis in melanoma metastasis. **a** Representative of four independent experiments. **b–f**, **h**, **j–k** Representative of three independent experiments. **g**, **i** Representative of two independent experiments. **b**, **d**, **e**, **h**, **i** Bars and error bars represent mean ± s.e.m. *N* = 3 (representative of three independent experiments). **a**, **c**, **f**, **g** Square brackets depict KAI1 molecular weights. **j** Square bracket depicts all forms of gp78, including endogenous and mCherry-tagged gp78.

including melanoma[38]. To further validate the role of KDELR3 on KAI1 protein regulation, we exogenously expressed KAI1 protein in 1205Lu metastatic melanoma cells (in which endogenous KAI1 expression is relatively low) and co-expressed KDELR3-001, KDELR3-002, or a vector control. Corroborating our initial findings, we found that increased KDELR3 expression resulted in dramatically reduced KAI1 protein levels (Fig. 5c), which could not be accounted for by KAI1 transcriptional changes (Fig. 5d, e). KAI1 protein glycosylation pattern was impacted reciprocally by knockdown and overexpression experiments, supporting the notion that KAI1 post-translational modification pathways are regulated by KDELR3, including an upregulation of a high-molecular-weight band in KDELR3-knockdown cells (Fig. 5f, red arrow) that we showed corresponds to a highly glycosylated form of KAI1 (Fig. 5g). Glycosylated KAI1 has been linked to inhibition of cell motility and promotion of cell death[39], and has been shown to influence N-cadherin clustering and bone metastasis in acute myeloid leukemia[40].

Owing to our protein expression data, we hypothesized that KDELR3 regulates KAI1 protein degradation. We asked if KDELR3 regulates expression of the E3 ubiquitin ligase known to target KAI1, gp78/autocrine motility factor receptor (AMFR)[20,41], hereafter referred to as gp78. Although we saw no significant alterations in gp78 protein or RNA expression following KDELR3 knockdown (Fig. 5f, h), we did observe a threefold increase in KDELR3 transcription following gp78/AMFR knockdown, suggestive of a functional link between the two proteins (Fig. 5i). We identified a previously undescribed interaction between KDELR3 and gp78, which was supported by evidence of co-localization (Fig. 5j, k; Supplementary Fig. 9a). Interestingly, gp78 was first identified as a motility factor associated with metastasis in several cancers[42], including melanoma. We asked if the KDELR3–gp78 interaction impacted its function. We reasoned that gp78 ubiquitin ligase substrates would be upregulated following gp78 knockdown, as these proteins would not be targeted for degradation; however, not all upregulated proteins identified will be direct gp78 substrates. Quantitative mass spectrometry was used to analyze whole-cell lysates of gp78 (AMFR)-knockdown or KDELR3-knockdown cells compared with non-targeting controls. We could confirm that 43–57% of upregulated proteins matched between the gp78- and KDELR3-knockdown groups. GSEA showed that the top-scoring, upregulated pathways (FDR <0.05) for both groups using GO term enrichment were those associated with the ER (Supplementary Tables 3 and 4). This result suggests that both gp78 and KDELR3 act within similar cellular pathways and support a role for KDELR3 in gp78 function, highlighting at least one mechanism through which KDELR3 can influence metastasis at the post-translational level. Since gp78 is a ubiquitin ligase known to function in ERAD, our data link KDELR3 to ERAD regulation. In summary, our work implicates KDELR3 in glycosylation of the metastasis suppressor, KAI1, and in its degradation through gp78 (and likely other ERAD effectors), thereby providing a mechanism for KDELR3 influence on the metastatic phenotype (Fig. 5l).

**KDELR3 correlates with late-stage metastasis and poor prognosis in melanoma patients**. To assess how KDELR3 contributes to melanoma progression in patients, we utilized multiple melanoma patient databases, TCGA[30,43] and Gene Expression Omnibus (GEO; GSE8401[25], GSE19234[27]). We found increased expression of the KDELR3-001 transcript, but interestingly not the alternate transcript, KDELR3-002, in late-stage (stages III and IV) metastatic melanoma patients compared with early-stage (stages I and II) melanoma patients (Fig. 6a), consistent with a role for KDELR3 in melanoma progression. Metastatic melanoma patients with KDELR3 copy number amplifications demonstrated

reduced survival relative to patients without such alterations (Supplementary Fig. 9b). We next assessed melanoma patient survival using KDELR3 expression as a prognostic marker (GEO[25,27]). High KDELR3-expressing late-stage metastatic melanomas showed statistically significant association with poor patient outcome, whereas KDELR3 expression levels in early-stage primary tumor samples did not (Fig. 6b, c). Taken together, these data strongly support a role for KDELR3 in the advancement of late-stage metastatic melanoma and implicate KDELR3 as a bona fide MetDev gene.

**KDELR3 and KDELR1 knockdown have opposing effects on lung colonization**. KDELR3 is the only KDELR family member we identified as a MetDev gene. Virtually all melanoma cell lines in the NCI60 were characterized by elevated KDELR3 expression, but reduced or unchanged expression of KDELR1 and KDELR2, respectively (Fig. 2b). We therefore posited that different KDELR members have different functions in melanoma etiology/progression. To address this, we asked which pathways were co-regulated with KDELR1 expression and if these were the same as or different from KDELR3-regulated pathways. GSEA analysis (FDR < 0.0001) of KDELR1 co-expressed genes in TCGA skin cutaneous melanoma patients (cBioPortal)[30,31] revealed a strong enrichment of mitochondrial, metabolic, and protein synthesis pathways (top 10 GO term enrichment, Fig. 7a), which differed from the most enriched pathways in KDELR3-co-expressed genes that consisted predominantly of ECM, trafficking, and ERSR pathways (top 10 GO term enrichment, Fig. 7b). Moreover, knockdown of KDELR3/KDELR1 did not consistently alter expression of each other, suggesting that expression of these genes is not intrinsically linked (Supplementary Fig. 9c, d). These data intimate that KDELR1 and KDELR3 play different roles in melanoma progression. To test this, we compared the behavior of KDELR3- and KDELR1-knockdown cells using experimental metastasis assays. Notably, in contrast to KDELR3 knockdown, which predictably diminished metastasis, KDELR1 knockdown actually increased metastasis, suggesting that KDELR1 contributes in a very different way to melanoma etiology and can function as a metastasis suppressor (Fig. 7c, d). Moreover, analysis of KDELR1 expression in skin cutaneous melanoma patients (TCGA) showed, unlike KDELR3, no significant difference between early- and late-stage metastatic melanoma patients (Fig. 7e). These data demonstrate that despite assumed redundancy between KDELR family members, KDELR3 and KDELR1 must have distinct roles, at least with respect to metastatic competence.

## Discussion

Here we propose that metastatic cancer cells exploit innate pathways that are hardwired within their cellular lineage to ensure proper development. These pathways, quieted in the differentiated cell, can be reactivated under pathologic conditions. The genetic/epigenetic reactivation of pathways that allow embryonic melanocytes to migrate, invade, and colonize would represent an efficient strategy for melanoma cells to successfully metastasize. Here we employed a GEM model to identify, at the transcriptome level, a set of genes that are upregulated during melanocyte development and find that these are enriched in genes that are specific for progression of late-stage disease. We functionally validated four out of four genes tested, demonstrating the value of our dataset and supporting our hypothesis. We anticipate that other genes that passed our filtering criteria will ultimately prove to be functionally relevant and deserving of further analysis in future studies.

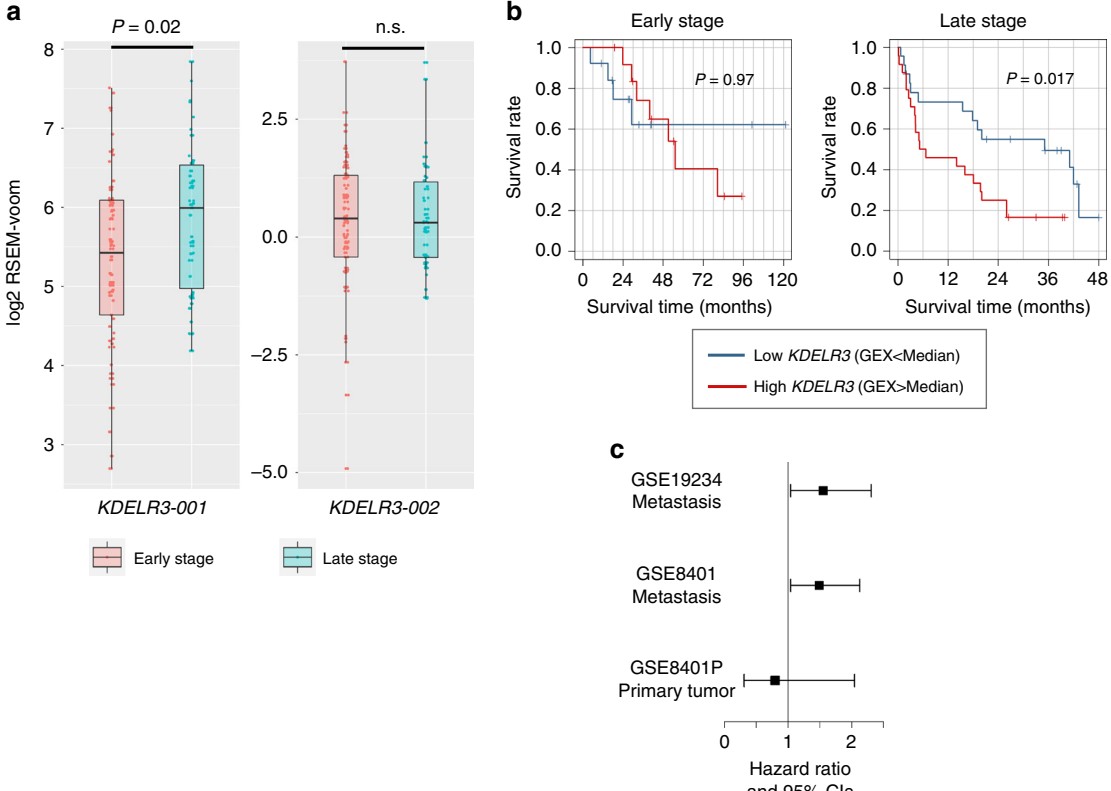

**Fig. 6 KDELR3 expression correlates with advanced metastatic disease in patients. a** *KDELR3-001* and *KDELR3-002* patient expression data. Empirical Bayes moderated *t*-statistic (unpaired two-tailed test); *KDELR3-001*, ENST00000216014, $P = 0.0202$, $t = 2.36$, d.f. $= 102.17$; *KDELR3-002*, ENST00000409006, $P = 0.87$, $t = 0.16$, d.f. $= 102.17$. Boxplots of patient expression data from TCGA-SKCM dataset, depicting the 25th, 50th (median), and 75th percentile, and extreme values of the transcript expression. "Early" stage (stages I/II, $N = 62$ patients). "Late" stage (stages III/IV, N = 39 patients). n.s., not significant. **b** Kaplan–Meier estimated survival curves according to *KDELR3* expression in early-stage (GSE8401; $n = 27$, stages I/II) and late-stage (GSE8401; $n = 47$, stages III/IV) melanomas. Log-rank test. **c** Association of *KDELR3* expression and survival in metastatic melanoma (GSE19234, $n = 38$; GSE8401, $n = 47$, stages III/IV); HR = 1.62 ($P = 0.028$) and HR = 1.49 ($P = 0.032$) for GSE19234 and GSE8401, respectively. No significant association was found in the primary tumors (GSE8401, $n = 27$, stages I/II); HR = 0.76 ($P = 0.509$). Cox regression model was used to test the association.

We report a mechanistic analysis of our top hit and melanoblast gene, *KDELR3*, a member of the KDEL receptor family. *KDELR3* has neither been previously associated with cutaneous melanoma metastasis nor investigated in depth in the literature. Differences between KDELRs have been cited in the literature, but the main focus has been the role of KDELR1[17,18,44,45]. All three KDELR family members have been shown to mediate retrograde transport of proteins containing a C-terminus KDEL-like motif[17]. KDELRs typically reside in the *cis*-Golgi; however, tagged KDELRs are known to localize in both the *cis*- and *trans*-Golgi, which is consistent with our results[46]. Upon interaction with KDEL-like motif-containing proteins, KDELRs facilitate transport from the Golgi apparatus back to the ER via COPI vesicles[47]. When this system fails, KDEL-like motif-containing proteins have been shown to be secreted out of the cell[17]. Our data demonstrating reduced BiP protein in stable *KDELR3*-knockdown cells suggest that BiP is a genuine substrate for KDELR3 retrograde trafficking, and that without *KDELR3* expression melanoma cells are unable to maintain normal BiP levels. KDELRs appear to differ in the substrates that they preferentially transport, suggesting that they have distinct roles within the cell[17]. How preferential substrate binding of KDELRs may affect cellular biology or disease etiology is largely unknown.

We show that distinct KDELRs mediate dramatically different experimental metastasis phenotypes. We demonstrate that the embryonic melanoblast gene, *KDELR3*, is a metastasis enhancer in both mouse and human melanoma cells, whereas *KDELR1* suppresses metastasis, despite having extensive homology and similar retrograde-trafficking functions. Our data provide a different perspective when interpreting existing KDELR literature and present a dichotomy between *KDELR3* and *KDELR1* metastasis phenotypes that could be leveraged in future studies to understand how these retrograde-trafficking receptors function in disease. Moreover, Trychta et al.[17] have reported tissue-specific KDELR expression patterns in rats, implying that different KDELRs may have lineage-specific roles. Our study allows assessment of *KDELR* expression in melanocyte development, revealing a specific role for *KDELR3*.

The KDEL receptors are intrinsically linked to ER stress and proteostasis. KDELR retrograde-trafficking substrates include protein chaperones, protein-folding chaperones, and protein-folding enzymes, enzymes that target proteins for degradation, and glycosylation enzymes[17]. Cumulatively, these protein substrates help maintain correct protein processing, and regulate cellular response to ER stress[18]. However, the role of ERSR in tumor progression has been much debated[48]. The success of proteasome inhibitors in the treatment of multiple myeloma patients[49], as well as provocative data linking ER stress pathways to vemurafenib-resistant melanoma and immunotherapy sensitization, suggest that UPR/ERAD biology could be harnessed for treating metastatic melanoma[50]. Our analysis implicates both

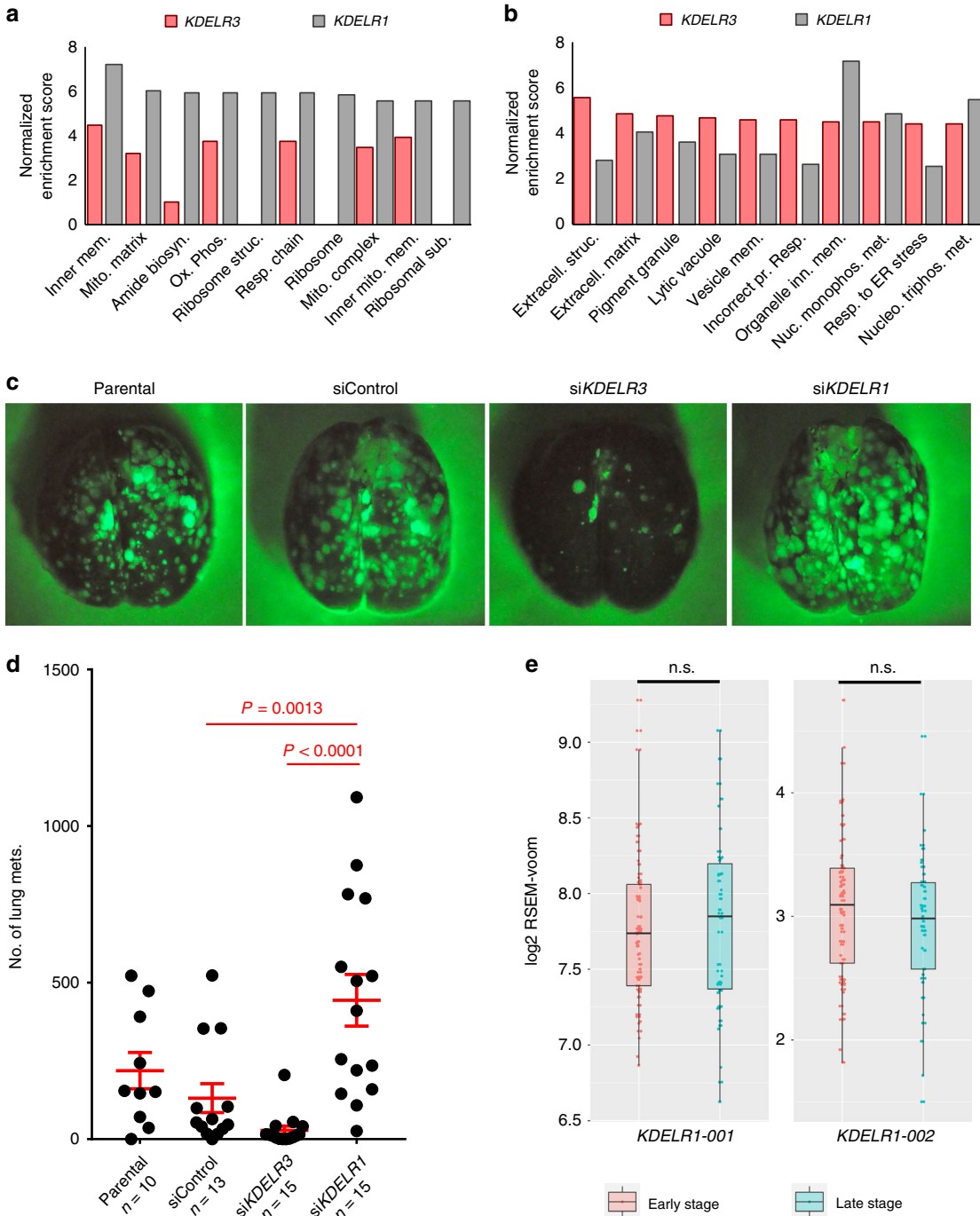

**Fig. 7 KDELR1 knockdown increases lung colonization in tail vein metastasis assays. a** GSEA of gene co-expression within skin cutaneous melanoma patients of the TCGA ($n = 479$). Top 10 KDELR1-associated GO pathways represented, FDR <0.0001. GO pathways in order: Organelle inner membrane; Mitochondrial matrix; Amide biosynthetic process; Oxidative phosphorylation; Structural constituent of ribosome; Respiratory chain; Ribosome; Mitochondrial protein complex; Inner mitochondrial membrane protein complex; Ribosomal subunit. **b** GSEA of gene co-expression within skin cutaneous melanoma patients of the TCGA ($n = 479$). Top 10 KDELR3-associated GO pathways represented, FDR <0.0001. GO pathways in order: Extracellular structure organization; Extracellular matrix; Pigment granule; Lytic vacuole; Vesicle membrane; Response to topologically incorrect protein; Organelle inner membrane; Nucleoside monophosphate metabolic process; Response to endoplasmic reticulum stress; Nucleoside triphosphate metabolic process. **c, d** Tail vein metastasis of KDELR1 siRNA-mediated knockdown ($N = 15$) human 1205Lu cells transduced with Ferh-luc-GFP. Parental ($N = 10$), siControl ($N = 13$), and siKDELR3 ($N = 15$) were used as controls. Images of whole mouse lung were taken at 1× magnification. ANOVA with Tukey's multiple comparison test; siControl vs. siKDELR1, $P = 0.0013$ [mean difference (95% CI): −312.5 (−521.7, −103.4)]; siKDELR3 vs. siKDELR1, $P < 0.0001$ [mean difference (95% CI): −415.3 (−616.8, −213.7)]. d.f. = 49, $F = 10.8$. CI, confidence interval of differences. Line and error bars represent mean ± s.e.m. **e** KDELR1-001 and KDELR1-002 patient expression data. Empirical Bayes moderated $t$-statistic (unpaired two-tailed test); KDELR1-001, ENST00000330720, $P = 0.73$, $t = 0.35$, d.f. = 102.17; KDELR1-002, ENST00000597017, $P = 0.39$, $t = −0.86$, d.f. = 102.17. Boxplots of patient expression data from TCGA-SKCM dataset, depicting the 25th, 50th (median), and 75th percentile, and extreme values of the transcript expression. "Early" stage (stages I/II, $N = 62$ patients). "Late" stage (stages III/IV, $N = 39$ patients). n.s., not significant.

UPR and degradation pathways of the ERSR as acting downstream of KDELR3. We show that *KDELR3* expression is critical for adaptation of melanoma cells to ER stress and provides evidence that PERK–EIF2α expression and activation are regulated by changes in *KDELR3* expression levels. Activation of the PERK–EIF2α pathway is known to result in translational attenuation, a cellular mechanism to alleviate ER load, causing translational rewiring of cells and affecting metastasis[13,15,48,51,52], which may contribute to KDELR3's metastatic role.

We demonstrate that KDELR3 is a regulator of glycosylated KAI1, a tetraspanin glycoprotein with a well-documented metastasis suppressor role in tumors, including melanoma[20,37,38,53,54]. KAI1 functions at the cell membrane to mediate interactions between extracellular and intercellular signaling, which is key to its metastatic suppressor function. KAI1 glycosylation leads to changes in its membrane organization and therefore its ability to mediate this extracellular/intercellular signaling[39,40,55]. However, no studies have linked specific KAI1 glycosylated forms with its metastasis suppressor function in vivo. Our work notes specific glycosylated forms of KAI1 that are subject to KDELR3 regulation and associated with metastatic function. Future studies would benefit from determining how critical each of these forms are to KAI1's metastatic influence in vivo. Previously, KDELR1 was shown to mediate signaling and transcriptional networks[44], and at the protein level, in the relocation of lysosomes and modulation of autophagy[56]. However, KDELR3 was shown to be inactive in these processes. Here we link KDELR3 to post-translational modification (glycosylation) and degradation of the metastasis suppressor, KAI1. Our data insinuate an interaction with gp78, implicating ERAD in this process. This biology may be informative for developing therapeutics for *KDELR3*-high metastatic melanoma patients.

We here identify an enrichment of ECM organization and trafficking genes within our MetDev cohort, consistent with a known role for these in metastasis[57,58]. Further analysis of these genes/pathways may prove a rich resource to study metastasis biology. We found that two such genes, *KDELR3* and *P4HA2* (a collagen prolyl 4-hydroxylase involved in ECM remodeling and associated with worse clinical outcome in melanoma patients[58]), from our four-gene functional validation screen are tightly co-expressed in four independent mouse models and in human melanoma patients. This raises the possibility that expression of some genes within our MetDev cohort may be coordinated and/or networked to realize the complex and dynamic phenotypes exhibited by melanocytic cells during development and metastasis. Uncovering common upstream regulators of co-regulated genes could prove a powerful approach to treat metastatic melanoma as multiple pathways could be targeted simultaneously.

Here we exploit the mouse melanoblast transcriptome to generate a resource of melanoma metastasis genes. The success of this study supports the use of developmental models to uncover innate melanoma biology that may be at the root of melanoma's propensity to metastasize[2–11,59]. We anticipate that further exploration of *KDELR3* and other now-uncovered embryonic genes/pathways will facilitate the development of more effective treatment strategies for patients with advanced melanoma, and perhaps other tumor types. The field would further benefit from elucidation of the specific melanoblast cell characteristics/cell states that in fact contribute to metastasis. In summary, this work provides a resource of putative MetDev genes, enriched in genes that have functional roles in melanoma metastasis that may prove to be useful targets for designing more effective approaches to the treatment of melanoma patients.

## Methods

**Mouse models of melanoma.** Experimental metastasis studies were performed using a filtered, single-cell suspension in phosphate-buffered saline (PBS). A total of $9.44 \times 10^5$ (1205Lu) and $2 \times 10^5$ cells (B16) were injected in 100-μl volume into the tail vein of 6–8-week-old athymic NCr-*nu/nu* female mice (01B74, Frederick National Laboratory for Cancer Research) or C57BL/6N mice (Charles River, Frederick National Laboratory for Cancer Research), respectively. Lungs were removed from mice 4.5 weeks (1205Lu) or 24 days (C57BL/6N) post injection, and then perfused and fixed in 10% phosphate-buffered formalin (Fisher Scientific) for histology. Metastatic nodules were counted under a dissecting microscope. Images of whole mouse lung were taken using a Nikon D90 camera with AF Micro Nikkor 60mm 1:2.8 D lens (magnification 1×). Fluorescent images of GFP-positive metastatic lung nodules were taken using a Nikon D90 camera with AF Micro Nior 60mm 1:2.8 D lens (magnification 1×) and a GFP filter (Lot# BN 532-62) with a NIGHTSEA fluorescence viewing system (Electron Microscopy Sciences).

Tumor growth studies were performed by injecting $3.47 \times 10^5$ 1205Lu cells in a single-cell suspension subcutaneously into the flanks of 6–8-week-old athymic NCr-*nu/nu* mice (01B74, Frederick National Laboratory for Cancer Research). Tumor size was estimated using the formula: tumor volume (mm$^3$) = $4/3\pi \times$ (length/2) × (width/2) × height, where parameters were measured in millimeters.

Melanoblasts and melanocytes were isolated from the i*Dct*-GFP mouse model[8]. Embryonic development was timed based on the number of days post coitum. Pregnant females and newborn pups were placed on a doxycycline-enriched diet to activate expression of GFP.

Melanomas in Fig. 4a, b and Supplementary Fig. 4a were derived from the following four mouse melanoma models: M1, albino female C57BL/6 background, with *Braf*$^{CA/+}$; *Pten*$^{flox/+}$; *Cdkn2a*$^{flox/+}$; *Tyr*-Cre$^{ERT2}$-tg transgenic alleles. Ultraviolet (UV) was used as the tumor-inducing carcinogen; M1 mice were treated on postnatal day 3. M2, C57BL/6 female background, with *Braf*$^{CA/+}$; *Cdkn2a*$^{flox/+}$; *Tyr*-Cre$^{ERT2}$-tg; *Hgf*-tg transgenic alleles. UV was used as the tumor-inducing carcinogen; M2 mice were treated at postnatal day 3. M3, C57BL/6 female background, *Cdk4*$^{R24C}$; *Hgf*-tg transgenic alleles. Dimethylbenz(*a*)anthracene was used as the tumor-inducing carcinogen; M3 mice were treated at postnatal day 3. M4, C57BL/6 male background, with *Hgf*-tg transgenic allele. UV was used as the tumor-inducing carcinogen; M4 mice were treated at postnatal day 3.

**Isolation of melanoblasts and melanocytes.** FVB/N i*Dct*-GFP dams were fed doxycycline-fortified chow for the entire duration of gestation until collection of E15.5, E17.5, and P1 pups. Doxycycline was injected intraperitoneally at 80 μg/g body weight 24 h before collection of P7 pups. A single-cell suspension was generated from embryos and skin of newborn pups. Multiple litters were used for each developmental stage, and embryos/pups from each stage were pooled to ensure adequate numbers of GFP$^+$ cells. The head was removed to prevent collection of GFP-positive cells in the embryonic telencephalon, and melanocytes from the inner ear or from the retinal pigmented epithelium were discarded. Excess tissue was also removed. The spinal cord was kept intact as some melanoblasts still remain in the neural crest area. At E17.5, P1, and P7 stages, most melanocytes have reached the dermis; thus, only the skin was collected from these developmental stages. Back skin was immersed in a shallow layer of 1× PBS and subcutaneous fat was scraped off until skin appeared translucent. E15.5 was the youngest age at which assessment was done due to the necessity to capture sufficient cells for RNA sequencing (RNA-seq).

**Preparation of single-cell suspensions.** Tissue was minced and incubated for 30 min at 37 ℃ in digestion media containing RPMI-1640 (Gibco Life Technologies) with 200 U/ml Liberase TL (Roche Applied Science). Up to 1 g of tissue was digested per 5 ml of digestion media. Tissue was processed using a Medimachine (BD Biosciences) and sterile medicon units (BD Biosciences). Cells were extracted using 1.5–2 ml of RFD solution (24 ml of RPMI media, 6 ml of fetal bovine serum, and 300 μl of 5% DNase I) through a 20-ml syringe with 18-gauge needle. Collected cells were filtered through a 50-μm filter (BD Biosciences). This process was repeated until all the tissue was processed. Cells were spun at $300 \times g$ at 4 ℃ for 5 min and resuspended twice in a solution of 1% bovine serum albumin (BSA) in PBS and filtered through a 30-μm filter for sorting.

**Fluorescence-activated cell sorting.** i*Dct*-GFP-positive melanoblasts/melanocytes were sorted using the BD FACSAria IIu or BD FACS Vantage (BD Biosciences) systems. FACS DIVA software was used during cell sorting and the FlowJo software for analysis. Cells were initially identified on forward scatter (FSC) vs. side scatter (SSC). Single cells were identified using FSC and SSC pause width. Cell doublets were excluded from the analysis. Embryos of the same developmental age that were heterozygous for the TRE-H2B-GFP gene but lacked the *Dct*-rtTA gene were used as negative controls. Cells were sorted based on GFP expression and SSC-A. GFP-positive cells were identified using appropriate gates based on negative controls. Due to low sample cell number, reanalysis of sorted cells was not usually done, but representative post-sort analyses confirmed that presort purities of 0.74–0.75% were enriched to 98–99.5%.

**RNA isolation and RNA-seq.** Cells were lysed in 10-fold TRIzol reagent (w/v), phases were separated by the addition of 0.2× volume of chloroform, and the aqueous phase was combined with an equal volume of 70% ethanol and applied to a RNeasy Micro column (Qiagen) and processed as per the manufacturer's instructions. Paired-end sequencing libraries were prepared using 1 µg of purified RNA following the mRNA-seq Sample Prep Kit according to the manufacturer's instructions (Illumina). RNA-seq libraries were sequenced on two lanes each of an Illumina GAIIx Genome Analyzer to a minimum depth of 49 million reads. Sequence reads were aligned to the mm9 genome using the TopHat software (https://ccb.jhu.edu/software/tophat/index.shtml). Quantified fragments per kilobase of transcript per million mapped read (FPKM) values were generated using the Cufflinks software (http://cole-trapnell-lab.github.io/cufflinks/). The UCSC KnownGenes gene models were used for guided alignment and quantification.

**Analysis of MetDev gene expression in melanoma progression.** The significant (FDR = 0.1) differentially expressed genes from $t$ test comparing metastatic vs. primary tumors were intersected with the 467 genes to obtain 183 genes in which 79 of them were upregulated in the metastatic tumors. Among the 79 genes, 66 genes had expression data in the dataset GSE98394. The expression of the 66 genes were transformed to $z$-statistics. We then applied the hierarchical clustering to divide the 66 genes into two groups and defined the 66-gene signature by assigning the positive weight 1 to the genes in the group containing KDELR3 and the negative weight −1 to the genes in the other group. The scores in Supplementary Fig. 3b were computed by using the 66-gene signature.

**Analysis of MetDev genes in patient survival.** Based on the RNA-seq data for the samples E15, E17, P1, and P7, we used DE-seq2 to find differentially expressed genes comparing E15, E17 vs. P1, P7. We selected 467 upregulated genes with $q$ value <0.1 (based on glm model) and $\log_2$ fold change >1.5. We then used the GEO dataset GSE19234 to perform survival analysis using Cox proportional hazards model for each gene. We selected 43 genes that correlated with patient overall survival, with a $P$ value <0.1 and a hazard ratio (HR) >1. Figure 1c, d showed the heatmaps of the gene expression (using $z$ score) for the 467 and 43 genes, respectively. The sum of the total expression of the 43 genes forms the expression signature for prognosis prediction and the signature was tested on the new dataset GSE8401. Among the late-stage patients, the patients with high expression signature had significant poor survival compared with those with low expression ($P = 3.486e − 5$, log-rank test, Fig. 1e), while for the early-stage patients the two groups had no difference in survival (Fig. 1e, f).

**Gene filtration pipeline for functional analysis.** From our 467 identified melanoblast genes, we first filtered for only those genes whose P7 expression level was low (FPKM <2), reasoning that these would denote genes that truly had a unique role in melanoblast development compared with differentiated melanocytes. Next, we validated these by identifying the genes that are the intersect of the 467 genes with the differentially expressed genes from microarray expression data derived from our iDct-GFP model (E17.5 vs. P2 and P7)[19]. The microarray differential gene expression was identified using a linear regression model with a contrast to compare embryonic vs. postnatal stages and selected with a $q$ value <0.1. The intersect yielded 233 genes. We acknowledge that the microarray data are not a thorough representation of melanoblast/melanocyte development as our developmental cohort, and therefore we may incur false negatives; we deemed this acceptable, however, to shorten our list for experimental validation. Next, we filtered the list to 81 genes with $\log_2$ fold change >2.75, corresponding to a $P$ value <0.0003, indicating increased expression in melanoblasts vs. melanocytes. Finally, we reasoned that genes with the greatest expression at embryonic stages would likely be the most functionally relevant, and thus were selected for the top 20 greatest mean embryonic expression. Of these 20, we noted that 7 genes (Kdelr3, P4ha2, Gulp1, Dab2, Lum, Aspn, and Mfap5) were all associated with ECM or trafficking. Of these, we chose to test the three least studied genes in melanoma metastasis (Kdelr3, Dab2, and Gulp1) to uncover metastasis biology, and the one gene known to be prognostic of worse clinical outcomes in melanoma (P4ha2).

**Statistical analysis of KDELR3 expression in microarray data.** Mouse developing melanoblasts (E17.5, $n = 3$) and differentiated melanocytes (P2, $n = 3$) were isolated and RNA extracted for microarray analysis as previously described[19]. The raw data (GSE25164 and unpublished, probe IDs 1690129, 4920546) from Illumina mouseRef-8 v1.1 (GSM618249) expression beadchip were processed with variance stabilization transformation and quantile normalization as implemented in R lumi package (http://bioconductor.org/packages/release/bioc/html/lumi.html). Unpaired two-tailed $t$ test with Welch's correction was used to compare the mean expression of KDELR3 between the two developmental stages. As two probes for KDELR3 on the Illumina beadchip showed high positive correlation ($r = 0.987$), the average KDELR3 expression was analyzed.

**Analysis of TCGA skin cutaneous melanoma expression.** All patient samples were collected between 0 and 14 days after disease classification (101 patients). Processed level 3 RNA-seq by expectation-maximization values[60] was imported for melanoma patients from TCGA collection (TCGA-SKCM). Bioconductor edgeR (https://bioconductor.org/packages/release/bioc/html/edgeR.html) and limma (https://bioconductor.org/packages/release/bioc/html/limma.html) R packages were used for further processing and differential expression analysis. Transcripts with CPM (counts per million) >1 in at least 50% of the samples were retained and processed with trimmed mean of $M$ values (TMM) and voom normalization methods[61]. The empirical Bayes moderated $t$-statistic test[62] was applied to test the null hypothesis both for no difference in KDELR3 expression and for KDELR1 expression level between early- and late-stage melanoma patients. A $P$ value of 0.05 or less was considered statistically significant.

**Statistics and general methods.** All sample sizes were determined based on preliminary studies and prior knowledge of expected variability within assays. For animal studies, age-matched (6–8 weeks) female athymic NCr-*nu/nu* mice and C57BL/6 mice were randomly assigned to control and test the groups. Blinding was used to quantify lung metastasis counts. Where blinding was not used, data were analyzed using automated image analysis software when possible. All statistical tests used were deemed appropriate and met the assumptions required; when parametric tests were used, normal distribution was assumed. Where necessary unequal variance was corrected for, or if no correction was used, variation was assumed equal based on prior knowledge of the experimental assay. For Western blot analyses, all unprocessed and uncropped scans of the most important blots have been included (Supplementary Figs. 10–12).

All mouse experiments were performed in accordance with Animal Study Protocols approved by the Animal Care and Use Committee, NCI, National Institutes of Health. NCI is accredited by AAALACi and follows the Public Health Service Policy on the Care and Use of Laboratory Animals. Studies were carried out according to ASP#16-007 and LMB-042. All animals used in this research project were cared for and used humanely according to the following policies: The US Public Health Service Policy on Humane Care and Use of Animals (2015); the Guide for the Care and Use of Laboratory Animals (2011); the US Government Principles for Utilization and Care of Vertebrate Animals Used in Testing, Research, and Training (1985). The experimental records of animal studies in this project are maintained in a style consistent with ARRIVE guideline. Here we follow the guideline to report the results of animal studies in this paper.

**Melanoma and melanocyte cell lines.** All cell lines used in this paper were identified correctly as per the International Cell Line Authentication Committee register of Misidentified Cell Lines, versions 8.0 and 9 (NB. MDA-MB-435 and MDA N cell lines in NCI60 were correctly identified as melanoma-derived cell lines). All cell lines used in the experiments were screened for mycoplasma contamination and were tested negative for mycoplasma contamination. Cell lines were authenticated by examining their expression of melanoma markers using quantitative PCR (qPCR) and reverse transcription-PCR (RT-PCR) analyses and validating expression levels to those previously reported in the published data. Human melanoma cell lines (1205Lu, WM-46, SK-MEL-28, and 234 hTERT-sh_p16) were validated using human *DCT*, *SOX10*, *TYRP-1*, and *TYR* primers. Mouse melanoma cell line (B16) was validated using mouse *Mitf*, *Trp2*, and *Tyr* primers.

Human melanoma cells, 1205Lu and WM-46, were obtained from the Wistar Institute (courtesy of Meenhard Herlyn, Wistar Institute). 1205Lu cells were cultured in Tu2% media (as described by the Wistar Institute). WM-46 and SK-MEL-28 (ATCC, HTB-72) cells were cultured in 1× RPMI-1640, with 10% serum and 2 mM L-glutamine (Gibco Life Technologies). For WM-46 cells, flasks were coated with 0.1% gelatin (Stemcell). B16 mouse melanoma cells were obtained from Isaiah Fidler, MD Anderson Cancer Center[63]. Human 1205Lu cells were transduced with a high multiplicity of infection (MOI) of FerH-ffLuc-IRES-H2B-eGFP-expressing lentivirus (11346-M04-653, Frederick National Laboratory for Cancer Research, Proteomics Facility, courtesy of Dominic Esposito)[63]. GFP-expressing cells were sorted using FACS (BD FACSDiva 8.0.1, Flow Cytometry Core Facility, National Cancer Institute).

GIPZ™ lentiviral shRNA particles were obtained from Dharmacon™. KDELR3 shRNA (V3LHS_307898, gene target sequence: TGTGCCTATGTTACAGTGT) or non-silencing negative control (RHS4348) lentivirus was infected at both 34–43 transducing units (TU)/cell and also at 25 TU/cell for a separate experiment. Cells were selected and maintained in puromycin selection.

Wobble mutant cell lines were generated using the QuikChange II Site-Directed Mutagenesis Kit (Agilent Technologies). The KDELR3 shRNA recognition sequence was edited (t210c_c213a_t216c_t219c_a222c) from Myc-DDK-tagged KDELR3 transcript variant 1 construct (RC201571, OriGene). TOPO cloning was used to clone place this sequence into the Gateway cloning system and the pENTR L1/L2 plasmid was combined with C413-E19 pPol2 L4/R1 and pDEST-658 R4/R2 destination plasmids. Lentivirus was produced in the Protein Expression Laboratory, Leidos Biomedical Research Inc., and Frederick National Laboratory for Cancer Research. Cells previously transduced and selected with KDELR3 shRNA and non-targeting control shRNA (Dharmacon, see previous) were transduced with 32.2 infection units per cell; cells were transduced by spinoculation for 1 h at 1200 × $g$. Infected cells were selected using blasticidin.
Forward primer:
5′-GTAATGAAGGTGGTTTTTCTCCTCTGCGCATACGTCACCGTGTACA
TGATATATGGGAAATTCCG -3′.

Reverse primer:
5′-CGGAATTTCCCATATATCATGTACACGGTGACGTATGCGCAGAGG AGAAAAACCACCTTCATTAC-3′.

Human KDELR3 expression vector was cloned using KDELR3 (NM_006855) sequence (SC122762, Origene) into pDest-653 destination vector by the Protein Expression Laboratory, Leidos Biomedical Research Inc., and Frederick National Laboratory for Cancer Research (mPol2p >Kz-KDELR3-eGFP IRES> ffluc2, 16876-M02-653). Lentivirus was produced in the Protein Expression Laboratory, Leidos Biomedical Research Inc., and Frederick National Laboratory. Cells were infected using a high MOI and infected cells were selected using FACS for GFP expression. FACS sorting was done using the BD FACSAria IIu (BD Biosciences) or BD FACSAria Fusion (BD Biosciences) systems. The FACSDiva 8.0.1 software was used during cell sorting and the FlowJo software was used for analysis. Cells were initially identified on FSC vs. SSC. Single cells were identified using FSC and SSC pause width. Cell doublets were excluded from the analysis. Cells were sorted based on GFP expression and SSC-A. GFP-positive cells were identified using appropriate gates based on negative controls. Post-sort analyses confirmed enrichment of 80–94%. A representative gating strategy is depicted in Supplementary Fig. 13.

Human gp78/AMFR expression vector was cloned using AMFR (NM_001144) sequence (RG209639, Origene) into pDest-653 destination vector by the Protein Expression Laboratory, Leidos Biomedical Research Inc., and Frederick National Laboratory for Cancer Research (mPol2p >Hs.AMFR-mCherry, 19771-M01-653). Lentivirus was produced in the Protein Expression Laboratory, Leidos Biomedical Research Inc., and Frederick National Laboratory. Cells were infected using an MOI of 5 and 8.8. Infected cells were selected using FACS for mCherry expression. FACS was done using the BD FACSAria Fusion (BD Biosciences) systems. The FACSDiva 8.0.1 software was used during cell sorting and the FlowJo software was used for analysis. Cells were initially identified using FSC vs. SSC. Single cells were identified using FSC and SSC pause width. Cell doublets were excluded from the analysis. Cells were sorted based on mCherry expression and SSC-A. mCherry-positive cells were identified using appropriate gates based on negative controls. Post-sort analyses confirmed enrichment of 89–91%. A representative gating strategy is depicted in Supplementary Fig. 14.

Primary melanocytes, of which there were 234 (gifted by Meenhard Herlyn, Wistar Institute), were immortalized through a one-step infection with MSCV-pic2 retroviral vector that co-express the catalytic subunit of hTERT and a shRNA CDKN2A gene locus that knocks down both p16$^{INK4A35}$ and p14$^{ARF35}$ (hTERT-sh_p16). Plasmids were packaged in RetroPack PT67 Cell Line with TransIT-X2 transfection reagent (Mirus). Viral supernatant was harvested at 55 h post transfection and passed through a 0.45-μm SFCA filter. Cells were transduced by spinoculation at 1200 × g for 45 min and supplemented with 4 μg/ml polybrene (Sigma) prior to infection. Immortalized melanocytes (234 hTERT-sh_p16) were selected by geneticin and retained non-tumorigenic status, as determined by lack of growth in anchorage-independent growth assays. The 234 hTERT-sh_p16 cells were cultured in Cascade Biologics Medium 254 (Gibco Life Technologies, Cat# M-254-500), supplemented with PMA-Free Human Melanocyte Growth Supplement-2 (HMGS-2, Gibco Life Technologies).

**siRNA knockdown of gene expression.** For experimental metastasis assays, siRNA-knockdown experiments were performed 2 days prior to injection, as follows: siGENOME Human KDELR3 (11015) siRNA SMARTpool (M-012316-02-0010, Dharmacon™) for KDELR3 siRNA knockdown in human cell lines, and siGENOME Mouse Kdelr3 (105785) siRNA SMARTpool (M-052192-00-0005, Dharmacon™) for Kdelr3 knockdown, siGENOME Mouse P4ha2 (18452) siRNA SMARTpool (M-040403-00-0005, Dharmacon™) for P4ha2 knockdown, siGENOME Mouse Dab2 (13132) siRNA SMARTpool (M-050859-01-0005, Dharmacon™) for Dab2 knockdown, and siGENOME Mouse Gulp1 (70676) siRNA SMARTpool (M-064490-01-0005, Dharmacon™) for Gulp1 knockdown, in the mouse cell lines. For control knockdown, siGENOME Non-Targeting siRNA Pool #1 was used (D-001206-13-20, Dharmacon). For KDELR1 knockdown in human cells, siGENOME Human KDELR1 siRNA SMARTpool (M-019136-01-0005, Dharmacon™) was used. Gene knockdown was done following the manufacturer's instructions using DharmaFECT 1 Transfection Reagent (T-2001-02, Dharmacon). All other assays were performed using both the siGENOME siRNAs, including siGENOME Human AMFR siRNA SMARTpool (M-006522-01-0005, Dharmacon™) and ON-TARGET Plus SMARTpool siRNAs for human KDELR3 (L-012316-00-0005, Dharmacon™), human KDELR1 (L-019136-01-0005, Dharmacon™), and ON-TARGET plus™ Control Pool (Non-targeting control, D-001810-10-20, Dharmacon™). We used either the DharmaFECT 1 Transfection Reagent (T-2001-02, Dharmacon) or the Mirus TransIT-X2® (Mirus) system as per the manufacturer's instructions. The results were consistent between all the knockdown methodologies.

**Anchorage-independent growth assays.** In 6-well plates, 50,000 cells (B16 Kdelr3-knockdown/SK-MEL-28 KDELR3 overexpression), 15,000 cells (WM-46 KDELR3 knockdown), or 2000 cells (WM-46 KDELR3 rescue experiments) were plated in 0.4% Bacto™ Agar (Becton, Dickinson and Company) in 1× RPMI-1640 (Gibco Life Technologies) solution over a layer of 0.5% Agar-RPMI. Media were replenished twice weekly, and cell growth was assessed at 4 weeks post plating. Wells were fixed in 10% methanol/10% acetic acid fixation solution with

subsequent staining using 0.01% crystal violet staining (Sigma-Aldrich) dissolved in 10% methanol solution. Colonies were analyzed under a dissecting microscope, and by imaging (Alpha Innotech imager) with subsequent analysis (Fluorchem HD2 software).

**Confocal imaging of iDct-GFP embryos.** Embryos were imaged using a Carl Zeiss laser scanning confocal microscope (LSM) 710, with an EC-Plan-Neofluar ×5 magnification 0.16 NA, air objective lens. The detector was a photomultiplier tube. The excitation was with a 488-nm laser and the emitted light was detected between 493 and 634 nm. The confocal pinhole was set to 69 μm. The acquisition software was Zen Black. The pixel resolution was set to 2.768 μm in X and Y, and 38.878 μm in Z, depth is 8 bit. Imaging was at room temperature. Using the Zen Black software, the image in (c) was noise reduced by application of a median filter with a kernel size of 5 × 5 × 3 (X × Y × Z) pixels. Then, the noise-reduced image was maximum intensity projected and a linear LUT (i.e., γ was 1.0) was used for displaying the image. Pseudocoloring was not used.

**Fluorescence imaging of iDct-GFP pups.** Maestro GNIR-FLEX fluorescence scanner (PerkinElmer Inc., Waltham, MA) was utilized to image the pups. Multispectral GFP images were captured (excitation filter: 457 ± 23 nm; emission filter: 490-nm long-pass liquid crystal tunable filter) by scanning through 500–720 nm at the step size of 10 nm. Raw images (without spectral unmixing) were used in the paper. Image acquisition and visualization were performed using the Maestro software 2.10.0 (PerkinElmer Inc., Waltham, MA).

**Fluorescence imaging of gp78 and KDELR3 co-localization.** 1205Lu human melanoma cells transduced with mPol2p> Hs.AMFR-mCherry and mPol2p> Hs. KDELR3-GFP were plated on chamber slides (Nunc Lab-Tek) coated with 0.1% gelatin (Stemcell) and imaged by confocal microscopy. Images were acquired on a Zeiss LSM880 using a ×63 plan-apochromat (NA 1.4) oil immersion objective lens. The 488- and 594-nm laser lines were used for excitation and fluorescence emission windows were set at 490–552 and 600–735 nm for GFP and mCherry, respectively. Images were collected using a 0.9-μm optical slice thickness, a 0.26-μm x–y pixel size, 8-bit data depth, 0.85-μs pixel dwell time, and 2× image frame averaging. A bright-field image was collected using the T-PMT detector during fluorescence image acquisition using the 488-nm laser.

**Immunohistochemistry and immunofluorescence staining.** Formalin-fixed paraffin-embedded (FFPE) immunofluorescence (IF) of iDct-GFP mouse skin sections was performed using heat-induced epitope retrieval (HIER) in Target retrieval buffer, Citrate pH 6 (Dako S1699) for 7 min in an immunohistochemistry microwave, followed by 15 min of cooling on the bench. Blocking was done with a protein block (Dako X0909), followed by avidin and biotin blocking was used (15 min each, Vector Laboratories SP-2001). Overnight incubation (4 °C) was with 1:200 rabbit monoclonal KDELR3 (NBP1-00896, Novus Biological; 1DB_ID, 1DB-001-0000718990), followed by incubation with a biotinylated secondary antibody (Vector Laboratories) and a 30-min incubation with Vectastain ABC HRP Kit (Vector Laboratories, PK-4001). Slides were then incubated with tyramide-conjugated AlexaFluor 594 for 7 min (Molecular Probes T20948) and then diluted 1:100 in amplification buffer with 0.0015% H$_2$O$_2$, labeled with Hoechst 33342 (Thermo Fisher Scientific), and coverslipped with Vectashield hardset (Vector Laboratories H-1400) and imaged (as below). Slides were then stored overnight in PBS at 4 °C and coverslips were removed the next day. Target retrieval and blocking steps were repeated, and then sections were incubated overnight (4 °C) with 1:200 GFP (D5.1) XP® monoclonal antibody (mAb) (Cell Signaling Technology Cat#2956S, RRID:AB_1196615), followed by incubation with a horseradish peroxidase-conjugated anti-rabbit secondary antibody, followed by a 30-min incubation with Vectastain ABC HRP Kit. Samples were incubated with 1:100 AlexaFluor 488 tyramide reagent (Molecular Probes T20950) in an amplification buffer with 0.0015% H$_2$O$_2$. Hoechst 33342 (Thermo Scientific) was added and slides were coverslipped (Vectashield hardset). Sections were analyzed using a Zeiss LSM510 using a ×40 plan-apochromat (NA 1.3) oil immersion objective lens. The 364-, 488-, and 543-nm lasers were used for excitation and fluorescence emission filters of bandpass 435–485 nm, bandpass 505–550 nm, and longpass 560 nm for the three fluorescence labels: Hoechst 33342, AlexaFluor 488, and AlexaFluor 594, respectively. Images were collected using 0.9-μm optical slice thickness, a 0.440-μm x–y pixel size, 8-bit data depth, 1.06-μs pixel dwell time, and 4× image frame averaging. A bright-field image was collected using the T-PMT detector during fluorescence image acquisition using the 488-nm laser.

FFPE lung sections were incubated for 15 min in Target retrieval buffer, pH 6 (Dako), using HIER, and left for 15 min to cool; 1:50 KDELR3 (NBP1-00896, Novus Biological; 1DB_ID, 1DB-001-0000718990, Lot#CA36131)/1:400 KDEL receptor 3 (L95) polyclonal (Bioworld Technology Cat#BS3124, RRID: AB_1663176, Lot#CA36131), and 1:250 HLA A (Abcam Cat#ab52922, RRID: AB_881225) antibodies were incubated for 1 h at room temperature. Polymer detection was performed with ImmPRESS AP Reagent Kit, anti-rabbit Ig (Vector Laboratories). Chromagen staining was done using ImmPACT™ Vector Red Alkaline Phosphatase Substrate Kit (Vector Laboratories), as per the manufacturer's instructions. Sections were analyzed by a board-certified veterinary

pathologist using the color deconvolution v9 algorithm in the Aperio Image Scope v12.0.1.5027 software. Metastatic counts were generated with particle analysis in the ImageJ software.

For TMA screen studies: FFPE melanoma progression TMA sections[29] were used. Tissue microarray (TMA) was constructed from archival formalin-fixed, paraffin-embedded tissue blocks. Briefly, 1.0-mm-diameter tissue cores were arrayed on a recipient paraffin block using a tissue arrayer (Beecher Instruments, Silver Spring, MD) where a representative tumor area was carefully selected for each tumor from hematoxylin and eosin-stained section of a donor block. TMA blocks were cut into serial 5-μm-thick sections, heated for 2 h at 60 °C, and then deparaffinized in xylene and rehydrated through a series of graded alcohol to distilled water. FFPE melanoma progression TMA sections were heat retrieved with pH 6 citrate buffer in a pressurized chamber (Pascal, Dako) and cooled for 15 min. Endogenous peroxidase activity was blocked using a 3% solution of aqueous hydrogen peroxide and nonspecific binding with additional 2% milk block for KDELR3 (L95). Subsequently, primary antibody hybridization was done with the following: rabbit polyclonal anti-KDELR3 (Bioworld Technology Cat#BS3124, RRID:AB_1663176, Lot#CA36131; 1:1000, 30 min, room temperature); rabbit polyclonal anti-P4HA2 (Cat#13759-1-AP, RRID:AB_2156286, Proteintech; 1:1000, 30 min, room temperature); rabbit polyclonal anti-DAB2 (Cat#PA5-56005, RRID: AB_2640364, Lot#UH2817567A, Invitrogen, 1:2500, 30 min, room temperature). Antigen–antibody complexes were detected using a peroxidase-conjugated EnVision + Rabbit Polymer Detection System (Agilent Technologies, Santa Clara, CA, USA) and visualized using 3,3′-diaminobenzidine (DAB). Slides were lightly counterstained with hematoxylin, dehydrated in ethanol, cleared in xylene, coverslipped, and evaluated by SMH at 20× under bright-field conditions.

For TMA KDELR3 validation: TMA (ME1004A, US Biomax) was processed by Histoserv Inc. (Germantown). Briefly, slides were deparaffinized in xylene and rehydrated through graded alcohol through to distilled water. Heat-mediated antigen retrieval was used. Sections were blocked with hydrogen peroxidase, followed by serum blocking prior to overnight incubation with 1:50 KDELR3 (NBP1-00896, Novus Biological; 1DB_ID, 1DB-001-0000718990, Lot#CA36131). The sections were incubated with a biotin-conjugated secondary antibody for 30 min. The tissue sections were visualized with AEC and counterstained with hematoxylin. The slides were finally dehydrated, cleared, and mounted. All incubations were carried out at room temperature and TBST was used as washing buffer. Sections were analyzed by a board-certified veterinary pathologist using the color deconvolution v9 algorithm in the Aperio Image Scope v12.0.1.5027 software. $H$ score was calculated by Aperio. $H = 1$ (% weak positive), +2 (% medium positive), and +3 (% strong positive) with a maximum score of 300.

For IF study of KDELR3 localization to the golgi: 1205Lu cells stably transduced with FLAG-tagged KDELR3-001 (ENST00000216014) were plated on glass coverslips coated with 0.1% gelatin (Stemcell) and fixed with 4% paraformaldehyde for 15 min. Cells were permeabilized in 0.1% Triton X-100 in PBS for 30 min and blocked in 4% BSA in 0.05% Triton X-100 in PBS for 10 min. Antibody incubation was for 1 h at room temperature with either 1:1000 anti-DDK (FLAG) clone 4C5 (OriGene, Cat#TA50011-100, RRID:AB_2622345, Lot#A031) and 1:100 anti-Golgin-97 (D8P2K) mAb (Cell Signaling, Cat#13192, RRID:AB_2798144, Lot#1), or 1:1000 Anti-DDK (FLAG) clone 4C5 (OriGene, Cat#TA50011-100, RRID: AB_2622345, Lot#A031) and 1:3200 GM130 (D6B1) XP® mAb (Cell Signaling, Cat#12480, RRID:AB_2797933 Lot#3). Then, it was co-stained with AlexaFluor 488 and 594 antibodies for 30 min at room temperature. Coverslips were mounted using mounting medium with DAPI (Vectashield, H-1200) and analyzed by confocal microscopy. Images were collected with a Zeiss LSM780 Elyra microscope using confocal mode and a ×63 plan-apochromat (NA 1.4) oil immersion objective lens. The 405-, 488-, and 561-nm lasers were used for excitation and fluorescence emission windows were set at 415–478, 500–552, and 570–632, for the three fluorescence labels DAPI, AlexaFluor 488, and AlexaFluor 594, respectively. Images were collected using a 1.1-μm optical section thickness, a 0.13-μm $x$–$y$ pixel size, 0.540-μm $z$-step size, 8-bit data depth, 1.27-μs pixel dwell time, and 2× image frame averaging.

KDELR3 antibodies were further validated using co-expression with DDK-tagged KDELR3 and analyzed using confocal microscopy.

**Flow cytometry analysis of melanoma cells.** Cell viability was assessed using LIVE/DEAD™ Fixable Violet Dead Cell Stain Kit (Invitrogen, Life technologies); 3 days post siRNA knockdown of KDELR3 or non-targeting control, cells were fixed and stained as per the manufacturer's instructions. When indicated, cells were treated with dimethyl sulfoxide vehicle control or 2.5 μg/ml tunicamycin 18 h before fixation. Cell staining was analyzed using the BD FACS CANTO II (BD BioSciences) system and the FlowJo v10 software. Cells were initially identified on FSC vs. SSC. Single cells were identified using FSC and SSC pause width. Positive staining was determined using appropriate gates based on unstained controls. A representative gating strategy for these experiments is exemplified in Supplementary Fig. 8a.

Cell cycle analysis was performed using incubation of live cells with 10 μM 5-bromo-2′-deoxyuridine (BrdU) for 45 min (1205Lu) or 90 min (WM-46). Cells were fixed dropwise with 100% ethanol to a final concentration of 70% ethanol at 4 °C. Cells were resuspended in 0.5 mg/ml RNase A (37 °C), and permeabilized with a solution of 5 M HCl 0.5% Triton X-100 in dH₂O for 20 min. Cells were incubated

with 1:200 BrdU antibody (Cell Signaling Technology Cat#5292S, RRID: AB_10548898, Lot#3) and stained with either 1:200 AlexaFluor 647 (1205Lu cells, Invitrogen) or 1:200 AlexaFluor 488 (WM-46 cells, Invitrogen), and then co-stained with 40 μg/ml propidium iodide (PI) solution. Flow cytometry was performed using the BD FACS CANTO II (BD BioSciences) system and analyses were performed in FlowJo v10. Data are displayed as pseudocolor plots. Cells were initially identified in FSC vs. SSC. Single cells were identified using PI pause width. Positive staining was determined using appropriate gates based on unstained controls.

**Reverse transcription and RT-PCR analysis of XBP1 splicing.** RNA was isolated using the RNeasy Kit (Qiagen). In some cases, cultured cells were homogenized using TRIzol® reagent (Ambion™) followed by vigorous agitation in chloroform, and then spun at $12,000 \times g$, 15 min (4 °C). The upper aqueous phase was utilized for RNA extraction using the RNeasy Mini Kit (Qiagen). Reverse transcription was carried out using the ImProm-II™ Reverse Transcription System (Promega) using Oligo (dT)20 oligonucleotides for poly-A tail detection, or by using the iScript cDNA Synthesisi Kit (BioRad). RT-PCR analysis of XBP1 splicing was carried out using XBP1 F, 5′-GGAGTTAAGACAGCGCTTGGGGA-3′ and XBP1 R, 5′-TGT TCTGGAGGGGTGACAACTGGG-3′ oligonucleotides and GoTaq® Green Master Mix (Promega), using a 58 °C annealing temperature for 25 cycles. The reaction yields a 164-bp band (XBP1 unspliced) and a 138-bp band (XBP1 spliced). GAPDH loading control: GAPDH F, 5′-GGATGATGTTCTGGAGAGCC-3′ and GAPDH R, 5′-CATCACCATCTTCCAGGAGC-3′.

**Real-time quantitative PCR analysis of gene expression.** SYBR Green dyes were used to run the qPCR: GoTaq® qPCR Master Mix (Promega) with the addition of CXR dye, or VeriQuest SYBR Green qPCR Master Mix (2×) (Affymetrix) or KAPA SYBR FAST qPCR Master Mix (Kapabiosystems). Reactions were carried out according to the manufacturer's guidelines on a 7900HT Fast Real-Time PCR system (Applied Biosystems) using the SDS 2.4 software; 57 °C/60 °C annealing temperatures and 40 cycles were used. Oligonucleotides designed to detect cDNA of the 18S rRNA were used as a loading control for human cDNA: 18S-F, 5′-CTTA GAGGGACAAGTGGCG-3′ and 18S R, 5′-ACGCTGAGCCAGTCAGTGTA-3′. Gapdh loading control was used for qPCR of mouse cDNA: Gapdh F, 5′-CTGG AGAAACCTGCCAAGTA and Gapdh R, 5′-TGTTGCTGTAGCCGTATTCA-3′. Individual human genes tested: KDELR3 F, 5′-TCCCAGTCATTGGCCTTTCC-3′ and KDELR3 R, 5′-CCAGTTAGCCAGGTAGAGTGC-3′; KDELR1 F, 5′-TCAAA GCTACTTACGATGGGAAC-3′ and KDELR1 R, 5′-ATTGACCAGGAACGCCA GAAT-3′; KDELR2 F, 5′-GCACTGGTCTTCACAACTCGT-3′ and KDELR2 R, 5′-AGATCAGGTACACTGTGGCATA-3′; KDELR3-001 F, 5′-TGACCAAATTG CAGTCGTGT-3′ and KDELR3-001 R, 5′-TCAGATTGGCATTGGAAGACT-3′; AMFR F, 5′-GGTTCTTAGTAAATACCGCTTGCT-3′ and AMFR R, 5′-TCTCAC TCACTCGAAGAGGGC-3′; CD82 F, 5′-TGTCCTGCAAACCTCCTCCA-3′ and CD82 R, 5′-CCATGAGCATAGTGACTGCC-3′.

**Exogenous expression studies.** For exogenous overexpression of CD82 and KDELR3 genes, the following expression plasmids were used: CD82 transcript variant 1 (NM_002231) Human Untagged Clone (Origene, CAT#SC324395), pCMV6-AC Tagged Cloning mammalian vector with non-tagged expression (Origene, CAT#PS100020), KDELR3 transcript variant 2 (NM_016657) Human Myc-DDK-tagged ORF Clone (Origene, CAT#RC216726), KDELR3 transcript variant 1 (NM_006855) Human Myc-DDK-tagged ORF Clone (Origene, CAT#RC201571), and pCMV6-Entry Tagged Cloning mammalian vector with C-terminal Myc-DDK Tag (Origene, CAT#PS100001). TransIT®-LT1 (Mirus Bio LLC). Expression plasmids were transfected into 1205Lu human metastatic melanoma cells. The manufacturer's guidelines were followed using a reagent: DNA ratio of 3 μl of TransIT®-LT1 Reagent per 1 μg of DNA.

**Western blot analysis of protein expression.** Cells were lysed with two methods: 1% Triton X-100 buffer (50 mM Tris, pH 7.5, 150 mM NaCl, 1% Triton X-100, 10 mM iodoacetamide, and phosphatase inhibitor cocktails 2 and 3) (Sigma-Aldrich) and cOmplete protease inhibitor cocktail (Roche), 50 μM MG132 or in RIPA lysis buffer (Sigma) with phosphatase inhibitor cocktails 2 and 3 (Sigma-Aldrich) and cOmplete™ protease inhibitor cocktail (Roche) as per the manufacturer's guidelines. Protein lysates were denatured in LDS sample buffer (Invitrogen) and sample-reducing agent containing dithiothreitol (DTT) (Invitrogen) at 70 °C for 10 min, and then run on a 4–12% Bis-Tris Nu-PAGE gel (Novex by Life Technologies) in MES SDS running buffer (Invitrogen). Nitrocellulose membranes were probed with the following antibodies: anti-PERK phospho (Ser713) antibody (BioLegend Cat#649402, RRID:AB_10640071, Lot#B203140), and Cell Signaling antibodies: anti-eIF2α (Cell Signaling Technology Cat#9722, RRID:AB_2230924, Lot#13), phospho-eIF2α (Ser51) (D9G8) XP™ Rabbit mAb (Cell Signaling Technology Cat#3398, RRID:AB_2096481, Lot#6), PERK (D11A8) rabbit mAb (Cell Signaling Technology Cat#5683S, RRID:AB_10831515, Lot#5), ATF-6 (D4Z8V) rabbit mAb (Cell Signaling Technology Cat#65880, Lot#1), BiP (C50B12) rabbit mAb (Cell Signaling Technology Cat#3177S, RRID:AB_2119845, Lot#8), and β-tubulin (9F3) rabbit mAb (Cell Signaling Technology Cat#2128, RRID:AB_823664, Lot#7). For immunoblotting, the rabbit mAb to CD82 (D7G6H) was used (Cell Signaling

Technology Cat#12439, RRIS:AB_2797915, Lot#1). Rabbit antibody to AMFR was used (Cell Signaling Technology Cat#9590, RRIS: AB_10860080). Anti-VINCULIN mouse mAb (Sigma-Aldrich, Cat#V9131, RRID:AB_477629). GAS1 rabbit polyclonal Ab (Origene Cat#AP51781PU-N, RRID:AB_11149892, Lot#SH08D402D), NME1/NDKA (NM23-H1) rabbit antibody (Cell Signaling Cat#3345, RRID: AB_2152700, Lot#1), Gelsolin (D9W8Y) rabbit mAb (Cell Signaling Cat#12953, RRID:AB_2632961, Lot#1), and BRMS1 rabbit polyclonal antibody (Invitrogen, Cat#PA5-78885, RRID:AB_2746001, Lot#U82788252).

**Mass spectrometry**. Cell lysates were extracted 4 days post siRNA knockdown of *KDELR3*, *AMFR*, or non-targeting control (siGENOME) using Dharmafect #1 transfection reagent. Cell lysates (250 μg each) were digested with trypsin using the filter-aided sample preparation protocol as previously described with minor modifications[64]. Lysates were first reduced by incubation with 10 mM DTT at 55 °C for 30 min. Each lysate was then diluted with 8 M urea in 100 mM Tris-HCl (pH 8.5) (UA) in a Microcon YM-10 filter unit and centrifuged at 14,000 × g for 30 min at 4 °C. The lysis buffer was exchanged again by washing with 200 μl of UA. The proteins were then alkylated with 50 mM iodoacetamide in UA, first incubated for 6 min at 25 °C, and then excess reagent was removed by centrifugation at 14,000 × g for 30 min at 4 °C. Proteins were then washed 3 × 100 μl of 8 M urea in 100 mM Tris-HCl (pH 8.0) (UB). The remaining urea was diluted to 1 M with 100 mM Tris-HCl, pH 8, and then the proteins were digested overnight at 37 °C with trypsin at an enzyme-to-protein ratio of 1:100 (w/w). Tryptic peptides were recovered from the filter by first centrifugation at 14,000 × g for 30 min at 4 °C, followed by washing of the filter with 50 μl of 0.5 M NaCl. The peptides were acidified and desalted on a C18 SepPak cartridge (Waters) and dried by vacuum concentration (Labconco). Samples analyzing the effect of *KDELR3* siRNA treatment alone were dimethyl labeled, as described, with the label being rotated between replicates[65]. Samples analyzing the effect of *KDELR3* or *AMFR* siRNA knockdown were quantitated using label-free methods. Dried peptides were fractionated by high pH reversed-phase spin columns (Thermo). The peptides from each fraction were lyophilized, and dried peptides were solubilized in 4% acetonitrile and 0.5% formic acid in water for mass spectrometry analysis. Each fraction of each sample was separated on a 75 μm × 15 cm, 2 μm Acclaim PepMap reverse- phase column (Thermo) using an UltiMate 3000 RSLCnano HPLC (Thermo) at a flow rate of 300 nL/min, followed by online analysis by tandem mass spectrometry using a Thermo Orbitrap Fusion mass spectrometer. Peptides were eluted into the mass spectrometer using a linear gradient from 96% mobile phase A (0.1% formic acid in water) to 35% mobile phase B (0.1% formic acid in acetonitrile) over 240 min. Parent full-scan mass spectra were collected in the Orbitrap mass analyzer set to acquire data at 120,000 full-width at half-maximum resolution; ions were then isolated in the quadrupole mass filter, fragmented within the HCD cell (HCD normalized energy 32%, stepped ±3%), and the product ions analyzed in the ion trap.

The mass spectrometry data were analyzed and either dimethyl labeling or label-free quantitation performed using MaxQuant version 1.5.7.4[66,67] with the following parameters: variable modifications—methionine oxidation and *N*-acetylation of protein N terminus; static modification—cysteine carbamidomethylation; first search was performed using 20 p.p.m. error and the main search 10 p.p.m.; maximum of two missed cleavages; protein and peptide FDR threshold of 0.01; min unique peptides 1; match between runs; label-free quantitation, with minimal ratio count 2. Proteins were identified using a Uniprot human database from November 2016 (20,072 entries). Statistical analysis was performed using Perseus version 1.5.6.0[68]. After removal of contaminant and reversed sequences, as well as proteins that were only quantified in one of the three replicate experiments, the quantitation values were base 2 logarithmized and non-assigned values were imputed from a normal distribution of the data. Statistically significant differences were assigned using a two-way *t* test with a *P* value cutoff of 0.05.

**Protein deglycosylation**. 1205Lu cells were transfected with control or *KDELR3* siRNAs as previously described in Methods. After 4 days cells were lysed in 1% Triton X-100 buffer (50 mM Tris, pH 7.5, 150 mM NaCl, 1% Triton X-100, 10 mM iodoacetamide, phosphatase inhibitor cocktails 2 and 3 (Sigma-Aldrich), and cOmplete protease inhibitor cocktail (Roche), 50 μM MG132). Lysates were diluted 1:2 with dH₂O to minimize lysis buffer effect. Ten microliters of deglycosylation mix buffer 2 (New England Biolabs) were added to 17 μg of protein (dissolved in water) for a 40 μl total volume. The solution was then heated at 75 °C for 10 min. After cooling, 5 μl Protein Deglycosylation Mix II (New England Biolabs) was mixed in gently. The reaction was incubated at room temperature for 30 min before being transferred to 37 °C for 1 hour. Reactions were analyzed by Nu-PAGE and immunoblotted with the rabbit mAb CD82 (D7G6H) (Cell Signaling Technology Cat#12439S).

**Co-immunoprecipitation**. Cells were lysed in 1% Triton X-100 buffer (50 mM Tris, pH 7.5, 150 mM NaCl, 1% Triton X-100, 10 mM iodoacetamide, phosphatase inhibitor cocktails 2 and 3 (Sigma-Aldrich), and cOmplete protease inhibitor cocktail (Roche), 50 μM MG132). Clarified lysates were precleared by incubation with Dynabeads Protein A (Thermo Fisher Scientific) at 4 °C for 30 min. Two milligrams of total protein lysates were immunoprecipitated with Dynabeads protein A–antibody complexes, using an anti-gp78 or anti-DDK antibody and their respective IgG

isotypes: rabbit IgG (BD Pharmingen) and mouse IgG (Santa Cruz). Incubation with rotation overnight at 4 °C was performed. Immunoprecipitates were washed five times with washing buffer (50 mM Tris, pH 7.5, 150 mM NaCl, and 0.1% Triton X-100) and resuspended in 50 μl of elution buffer containing washing buffer, Nu-PAGE LDS sample buffer, and Nu-PAGE sample-reducing agent, mixed as per the manufacturer's instructions (Invitrogen). Proteins were analyzed by Nu-PAGE and immunoblotted using an enhanced chemiluminescence (ECL) method. For immunoblotting anti-DDK antibody (Origene TA50011) or (Ab2) to amino acids 579–611 of gp78 was used; this antibody was previously described[20].

**Reagents**. Any unique reagents generated in this study are available (within reason) from authors upon request.

**Reporting summary**. Further information on research design is available in the Nature Research Reporting Summary linked to this article.

## Data availability
The iDct-GFP RNA-seq data have been deposited in the GEO database under the accession code "GSE140193". The iDct-GFP microarray data have been deposited in the GEO database under the accession code "GSE25164". The TCGA data referenced during the study are available in a public repository from the cBioPortal for Cancer Genomics website. The source data, code, and figures underlying Figs. 6a–c and 7e are provided as a Source Data file deposited in the Figshare database under accession code [https://figshare.com/s/c0346519a91b02161f35]. All the other data supporting the findings of this study are available within the article and its supplementary information files and from the corresponding author upon reasonable request. A reporting summary for this article is available as a Supplementary Information file.

## Code availability
R scripts used in this study are deposited under accession code [https://github.com/maxplee/paper/blob/master/kerrie.paper.script.r]. R scripts (R version 3.5.0) underlying Fig. 6a–c are deposited in the Figshare database under accession code [https://doi.org/10.6084/m9.figshare.c.4710005.v1]. Software citation: R Core Team (2018). R: A language and environment for statistical computing. R Foundation for Statistical Computing, Vienna, Austria. URL https://www.R-project.org/.

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

## Acknowledgements

This research was supported in part by the NCI Intramural Research Program of the NIH. P.J.M. was also supported in part by the NCI Director's Innovation Award. M.R.Z. was supported in part by the following grant: NIH/NCI K22CA163799. T.G. was supported in part by the HHMI Research Scholars Program, Howard Hughes Medical Institute. H.T.M. funded in part by the NIH Comparative Biomedical Scientist Training Program in partnership with the University of Maryland, College Park, and the National Cancer Institute. We would like to thank Dr. Meenhard Herlyn for providing melanoma cell lines included in this study, and Dr. Ji Luo for providing the hTERT_p16 plasmid. We are grateful to Cari Graff-Cherry for maintenance and care of mouse lines, Dr. Stephen Lockett for microscopy analysis, and Dr. Joe Kalen for imaging developing mice, and Drs. Dominic Esposito and Carissa Grose for lentiviral vector preparation. We

acknowledge Dr. Yves Pommier and William Reinhold for NCI60 data and CellMiner analyses. We thank Jennifer Dwyer, Shelley Hoover, and Bih-Rong Wei from the Molecular Pathology Unit for slide scanning and IVIS imaging. We acknowledge Leidos at the British Columbia Cancer Agency for.pngting us free RNA-seq services. We are also grateful to Dr. Lalage Wakefield for useful discussions.

## Author contributions

K.L.M. and G.M. wrote the paper. G.M., K.L.M., P.J.M., A.S., A.M.M., C.-P.D., A.M.W., Y.C.T., H.T.M., S.D., P.S.M. and L.M.J. participated in experimental/study design. K.L.M., P.J.M., A.S., A.M.M., H.T.M., T.G., Y.C.T., M.R.Z., E.P.-G., S.D., L.M.J., S.M.H., K.Y. and N.L.P. generated the experimental data. G.M., K.L.M., P.J.M., A.S., A.M.M., M.P.L., H.H.Y., H.T.M., Y.C.T., A.M.W., E.P.-G., C.-P.D., H.A., S.D. and P.S.M. contributed intellectually to the work. M.P.L., H.H.Y., A.M.M. and S.D. performed bioinformatic and statistical analysis of data.

## Competing interests

The authors declare no competing interests.
