## [Peer Review File · Nature Communications]

Reviewers' comments:

Reviewer #1 (Remarks to the Author):

The study by Marie and colleagues takes a highly novel approach to identify metastasis genes in melanoma. The authors identify genes that are as potential metastatic drivers by focusing on those expressed during development in a mouse model followed by cross species validation to human metastatic cancer. They identify a robust gene signature which is prognostic of metastatic melanoma in human patients. Following which they validate one of the genes -- KDELR3 - and show that its depletion impairs metastasis. The authors include an interesting and novel analyses of the mechanism of action of KDELR3.

This reviewer is hard pressed to identify substantive criticisms of this fine work. The manuscript is exceptionally well written, the data are clearly presented and convincing. The statistical analysis is complete and clear.

Furthermore, the concept of identifying metastasis-regulated genes based on developmental genes is highly novel and at the same time provides fundamental validation of long standing ideas about the relationship of developmental pathways and metastatic ones.

Thus, I anticipate that this fine study will be of considerable relevance to the readership of Nature Communications.

Reviewer #2 (Remarks to the Author):

The authors herein perform an analysis of murine melanoblast, murine melanocyte and human melanoma cell transcriptomes to identify genes critical for melanoma metastasis. The authors identify KDELR3 as serving a role in metastatic progression and show genetic silencing KDELR3 can reduce metastatic potential of cells. The manuscript data is spread amongst 10 figures, but should be condensed to ~6 full figures that are each more comprehensive and informative than they are at present. Below are the issues to be addressed:

- The authors compare murine melanocyte and Melanoblast transcriptomes to human melanoma transcriptomes. The authors should determine whether their findings are consistent with human melanocyte and Melanoblast transcriptomes.
- Figure 2: How many distinct murine melanocyte and murine Melanoblast cell lines were sequenced?
 - o Figure 2A would benefit from showing the melanoma cell transcriptomes alongside the melanocyte/melanoblast transcriptomes to assist the reader in interpreting the data
- Figure 1 in its current form is not informative and is rather a supplemental style figure
- The authors mention in Figure 2D and 2E that their signature does not predict patient outcome from primary melanoma lesions. Where were the melanoma cell lines they used derived from? All metastatic lines or a mix of primary and metastatic? This information should be provided and referenced to.
- This work would benefit from IHC validation in patient metastatic versus primary tumor tissue of the Melanoblast biomarkers.
- Figure 3: Authors should validate the expression of KDELR3 across a panel of human Melanoblast, melanocyte and melanoma cell lines side by side to demonstrate the shared expression in melanoma and Melanoblast. At least 3 cell lines for each cell type should be used. Antibodies for KDELR3 exist, why are they not used to demonstrate protein expression rather than mRNA?
- Figure 5: Have the authors tried using a spontaneous metastasis model to better demonstrate the metastasis progression importance of KDELR3?
 - o The authors state in Figure 6 that KDELR3 KD causes a >5-fold increase in cell death versus controls. This brings up the question whether the reduction in metastasis is just due to less viable cells being injected into the tail vein.
- Figure 6C: The authors should rerun and display the WBs on one continuous blot to demonstrate the increase in p-PERK and PERK following tunicamycin treatment versus untreated.
- Figure 7B: What is the rationale to determine whether KDELR3 and gp78 physically interact. The authors state this was previously unreported, but the authors provide no clear scientific rationale to check versus any other ER machinery.
- Without labeling some of the genes including KDELR3, Figure 10A in its current form is not informative
- Figure 10B has no y-Axis label
- The authors should demonstrate whether KDELR3 is critical for melanoblasts or melanocyte viability.

Reviewer #3 (Remarks to the Author):

Marie et al present an integrative study of transcriptomes of metastatic melanoma in a mouse model and human patients. They derived a candidate gene list common to mouse transcriptomes, human cell lines and patient data and associated the genes with pathway activity and patient survival characteristics. They pursued comprehensive functional validation of one candidate gene KDELR1 and propose it as a regulator of metastasis. I believe that the functional validation data are rather strong. I focused on the computational biology of this study and found a number of important limitations that need to be addressed.

1. the composition of the initial gene signature of 16 genes, the premise of this study, appears as an ad-hoc series of unjustified filters. There were 149 genes in the comparison of mouse melanoblasts and melanocytes with strong associations in the pathway level (eg. cancer evasion, $q=10^{-17}$; how many genes did that pathway include?), already indicating the strong relevance of the entire gene list. However, the authors also looked at NCI human cell lines, mouse qPCR and other factors to further pursue only 16/149 genes, of which 8/16 were found up-regulated in human metastatic melanomas (patient data). Why did they not look at 149 mouse genes directly in human patient data (TCGA)? Would their 16-gene list (and the candidate gene KDELR3) still rank highly if mouse transcriptomic data were more directly compared with patient transcriptomic data? Different processing pipelines, filtering steps and thresholds they used have their intrinsic biases and likely lead to false negatives in such a strongly filtered list.

2. The analysis of up-regulation of MetDev genes in MITF-low patients is problematic for largely the same reasons listed in point 1 above.

A. First, the authors apply arbitrary selection of top-3000 most expressed genes on page 13 (E15.5 and E17.5), combined with down-regulation at P1 and P7 at $P < 0.05$. Instead of selecting an arbitrary list of genes (top-3000, why not top 100 or 1000?), a principled analysis would compare embryonic and post-natal timepoints using a statistical model (limma/voom or edgeR).

B. Second, an FDR-adjusted p-value needs to be used for gene filtering instead of an unadjusted P-value.

C. Third, the authors use some arbitrary list of GO categories to filter their 3000 (?) genes. They will miss genes with limited GO annotations. Are the GO-annotated gene lists they use up-propagated (i.e., do these contain all genes in child GO terms of these selected terms?)

D. Fourth, in the same analysis and hidden in methods, top-15% most variable patient genes from TCGA were intersected with the above genes from mouse (after homology mapping). The result is a small list of 94 genes that almost appears to result from data tinkering rather than principled analysis. Why top-15% and not some other number?

In particular, once the authors find mouse genes different between embryonic and post-natal timepoints (eg limma/voom $FDR < 0.05$), it would be natural to use all these genes to perform hierarchical clustering and significance tests between phenotypic groups (MITF, keratin, immune).

3. The survival analysis does not appear to consider important clinical covariates, such as tumor stage, patient age, patient sex, tumor subtypes (<https://www.ncbi.nlm.nih.gov/pmc/articles/PMC4580370/>). This information is available in TCGA. Does the Cox survival model of their gene signature outperform another control Cox model where clinical variables are listed as predictors?

4. Several uses of incorrect gene correlation filter are apparent [Spearman's rank of > 0.3 and p-value < 0.0001]. Multiple testing correction (FDR) should be used at all times. E.g. top of page9, page13 in the manuscript.

5. Methods on selecting 149 genes (review point 1; found on page 20). "Genes differentially expressed during development were calculated by ≥ 5 -fold decrease of expression in P7 versus E15.5 (FPKM)". This five-fold increase is problematic because we do not know variation between replicates. The authors need to use a statistical model such as limma/voom or edgeR to determine significant changes between groups, fold-change alone is not sufficient. The method TREAT by the developers of Limma actually allows to combine fold-change filters with p-value filters if necessary for this analysis.

6. Analysis of TCGA data also seems to use non-adjusted P-values for filtering genes, FDR-adjusted values should be used instead (found on page 23).

7. Why did the authors use different RNA-seq processing methods (eg Limma/voom for TCGA; DEseq2 for NCI cell lines; apparently no model for mouse transcriptomics)? Similarly, different pathway analysis methods were used (ingenuity for mouse transcriptomics, Fisher's test with MSigDB for NCI cell lines). It would look much more reliable if a consistent series of steps were applied with standard statistical filtering criteria (e.g., $FDR < 0.05$).

REVIEWER 1

We thank Reviewer 1 for their positive comments about the novelty of the approach, the robust analyses of the mechanism of action of KDELR3 and the fundamental validation of long-standing ideas about the relational of developmental pathways and metastatic ones. We agree with this reviewer that this work will be of considerable relevance to the readership of *Nature Communications*.

REVIEWER 2

We thank Reviewer 2 for their suggestions to further our functional validation. We have addressed all major concerns. We have listed Reviewer 2's suggestions below and indicated how we have addressed each (*bold-italics*):

- The authors compare murine melanocyte and Melanoblast transcriptomes to human melanoma transcriptomes. The authors should determine whether their findings are consistent with human melanocyte and Melanoblast transcriptomes. *We would be delighted to generate this data if possible. However, while RNA-sequencing data of human epidermal melanocytes does exist, this is inconsequential if we cannot compare this to human embryonic melanoblast data. To our knowledge, there are no existing human embryonic melanoblast transcriptome datasets and it is not possible for us to generate a dataset such as this in the United States. This perfectly illustrates why leveraging mouse models is so important to answer these questions.*
- Figure 2: How many distinct murine melanocyte and murine Melanoblast cell lines were sequenced? *We apologize if this was unclear. Cell lines were not sequenced. We sequenced murine melanocytes and melanoblasts extracted directly from the mouse skin of iDct-GFP mice. Transcriptome data was generated from an amalgamation of multiple litters from each developmental stage, consisting of 6-10 pups each, thus ensuring comprehensive representation of all melanoblasts/melanocytes present. We have altered the text to make this clearer (Page 4 and 18).*
- Figure 2A would benefit from showing the melanoma cell transcriptomes alongside the melanocyte/melanoblast transcriptomes to assist the reader in interpreting the data. *We agree. See Supplementary Figure 3a-b.*
- Figure 1 in its current form is not informative and is rather a supplemental style figure. *We agree. Moved to Supplementary Figure 1.*
- The authors mention in Figure 2D and 2E that their signature does not predict patient outcome from primary melanoma lesions. Where were the melanoma cell lines they used derived from? All metastatic lines or a mix of primary and metastatic? This information should be provided and referenced to. *We apologize if this was not clear. These data refer to patient-derived samples, not cell lines. We have clarified this in the text (Page 5).*
- This work would benefit from IHC validation in patient metastatic versus primary tumor tissue of the Melanoblast biomarkers. *We thank the reviewer for this suggestion. IHC validation of protein candidates was undertaken in collaboration with Drs. Stephen Hewitt and Kris Ylaya of the*

Experimental Pathology Laboratory, Center for Cancer Research (CCR), NCI. The NCI melanoma progression Tumor MicroArray (TMA) panel was used and for the 3 available robust antibodies (KDEL3, P4HA2, DAB2) all proteins demonstrated a marked increase in protein expression with progression (Page 6 and Supplementary Fig. 3c-h). Additionally, at the RNA level, we confirmed that expression of candidates KDEL3, DAB2 and GULP1 increased significantly in correlation to melanoma progression alongside a cohort of MetDev genes (Page 6 and Supplementary Fig. 3a-b, 79 MetDev genes interrogated).

- Figure 3: Authors should validate the expression of KDEL3 across a panel of human Melanoblast, melanocyte and melanoma cell lines side by side to demonstrate the shared expression in melanoma and Melanoblast. At least 3 cell lines for each cell type should be used. Antibodies for KDEL3 exist, why are they not used to demonstrate protein expression rather than mRNA? *This would be a worthwhile analysis if feasible. However, human melanoblast cell lines do not exist/are not available in this country, therefore the comparison could only be between melanocyte and melanoma cell lines, which would not answer the reviewer's question. Antibodies for KDEL3 do exist and we have validated appropriate staining patterns at the IHC and IF level. However, for our lysis conditions and Western conditions, we were unable to find a robust antibody that we could validate at the Western blot level. Therefore, we decided our protein level studies would be far more accurate using exogenously expressed tagged protein. These protein data were consistent with our RNA data for KDEL3.*

- Figure 5: Have the authors tried using a spontaneous metastasis model to better demonstrate the metastasis progression importance of KDEL3? *We thank the reviewer for this suggestion. However, we elected to focus on the tail vein experimental metastasis mode, which assesses lung colonization specifically, over the spontaneous metastasis model, which in contrast is much more complex and requires additional steps such as dissemination from the primary tumor and intravasation prior to metastatic colonization. We reasoned that a metastatic colonization model would be far more relevant to the biology we uncovered. Critical to our reasoning was that during the embryonic time points we were studying (E15.5 and E17.5), melanoblasts were mainly colonizing the developing epidermis and the developing hair follicle.*

- The authors state in Figure 6 that KDEL3 KD causes a >5-fold increase in cell death versus controls. This brings up the question whether the reduction in metastasis is just due to less viable cells being injected into the tail vein. *We appreciate the reviewer's question. However, we believe this is not the case for three reasons: 1) Prior to injection, all dead (floating) cells were washed away and trypan blue staining confirmed the percentage live cells in the remaining pellet was indistinguishable between KDEL3 knockdown and controls. 2) Stable shRNA knockdown of KDEL3 does not have a cell viability phenotype at the time of injection and we still observe a difference in metastasis phenotypes between shControl and shKDEL3 knockdown cells (See Supplementary Figure 6c-f). 3) The difference in cell viability between siControl and siKDEL3 cells (without Endoplasmic Reticulum stress induction) is relatively small and therefore highly unlikely to contribute entirely to the dramatic difference between metastasis phenotypes.*

- Figure 6C: The authors should rerun and display the WBs on one continuous blot to demonstrate the increase in p-PERK and PERK following tunicamycin treatment versus untreated. *We agree: See Figure 4c.*

- Figure 7B: What is the rationale to determine whether KDEL3 and gp78 physically interact. The authors state this was previously unreported, but the authors provide no clear scientific rationale to check versus any other ER machinery. *We apologize that our rationale for determining that KDEL3 and gp78 physically interact was not clear. To better elucidate this point, we first have included metastasis suppressor screen data showing KDEL3 knockdown only impacts protein expression of KAI1, but not other known melanoma metastasis suppressors. And second, KAI1 protein is known to be regulated by gp78, and therein lay our rationale (See figure 5a).*

- Without labeling some of the genes including KDELR3, Figure 10A in its current form is not informative. *We thank the reviewer for this suggestion. Figure 10 has since been removed to simplify the presentation.*
- Figure 10B has no y-Axis label. *Figure 10 has since been removed to simplify the presentation.*
- The authors should demonstrate whether KDELR3 is critical for melanoblasts or melanocyte viability. *We are grateful to the author for this worthy suggestion. We provide new data showing that KDELR3 was only critical for metastatic melanoma viability, but not melanocyte viability (See Supplementary Figure 8a-b).*

REVIEWER 3

We are grateful to Reviewer 3 for all their suggestions and comments. Firstly, we would like to thank the reviewer for their comment, referring to KDELR3: “I believe that the functional validation data are rather strong”. Secondly, we are indebted to Reviewer 3 for their suggestions to update the computational analysis. We have now addressed all major concerns and the resulting improved computational methods have greatly strengthened the foundational analyses of our study. We have listed Reviewer 3’s suggestions below and indicated how we have addressed these (*bold-italics*):

1. The composition of the initial gene signature of 16 genes, the premise of this study, appears as an ad-hoc series of unjustified filters. There were 149 genes in the comparison of mouse melanoblasts and melanocytes with strong associations in the pathway level (eg. cancer evasion, $q=10^{-17}$; how many genes did that pathway include?), already indicating the strong relevance of the entire gene list. However, the authors also looked at NCI human cell lines, mouse qPCR and other factors to further pursue only 16/149 genes, of which 8/16 were found up-regulated in human metastatic melanomas (patient data). Why did they not look at 149 mouse genes directly in human patient data (TCGA)? Would their 16-gene list (and the candidate gene KDELR3) still rank highly if mouse transcriptomic data were more directly compared with patient transcriptomic data? Different processing pipelines, filtering steps and thresholds they used have their intrinsic biases and likely lead to false negatives in such a strongly filtered list. *We thank Reviewer 3 for these valuable comments and in response have collaborated with bioinformatician, Dr. Maxwell Lee, CCR, NCI to overhaul the computational analysis based on these suggestions. We have used a generalized linear model, implemented in the R package DESeq2, to identify 467 genes that are upregulated in melanoblasts and downregulated in melanocytes. This entire dataset was used in analysis of human patient data as requested. These analyses resulted in several important observations. 1) Independent analyses uncovered melanoblast genes that were correlated with advanced melanoma stage in two independent patient datasets, and expression levels of our candidate genes corroborated these findings - in particular, KDELR3 (See Page 5, Supplementary Fig. 3. a-b). 2) Melanoblast genes were found to be associated with poor progression in late stage metastatic melanoma patients in two independent patient datasets (See Page 5, Figure 1d-f). 3) Recognizing that, as Reviewer 3 suggested, different processing pipelines and thresholds among distinct datasets could have led to intrinsic biases, we selected a few key candidate genes for functional validation based solely on expression levels in melanoblasts versus melanocytes. Of the top 20 candidates, EMT and trafficking pathways were predominant. Four candidate genes were therefore selected that were upregulated in advanced melanoma stage and have been indicated in all analyses (See Pages 5-6, Fig. 1c-d, Supplementary Fig. 3 a-c). Strikingly, despite a complete overhaul in computational analysis, these 4 functionally validated genes were consistent with those validated in the previous manuscript, i.e., two independent analyses revealed the same candidates, adding further weight to their relevance in melanoma metastasis. Moreover, KDELR3 was validated throughout all*

analyses we undertook, thereby further supporting our rationale to select KDELR3 for follow up as a melanoma metastasis gene.

2. The analysis of up-regulation of MetDev genes in MITF-low patients is problematic for largely the same reasons listed in point 1 above.

A. First, the authors apply arbitrary selection of top-3000 most expressed genes on page 13 (E15.5 and E17.5), combined with down-regulation at P1 and P7 at $P < 0.05$. Instead of selecting an arbitrary list of genes (top-3000, why not top 100 or 1000?), a principled analysis would compare embryonic and post-natal timepoints using a statistical model (limma/voom or edgeR).

B. Second, an FDR-adjusted p-value needs to be used for gene filtering instead of an unadjusted P-value.

C. Third, the authors use some arbitrary list of GO categories to filter their 3000 (?) genes. They will miss genes with limited GO annotations. Are the GO-annotated gene lists they use up-propagated (i.e., do these contain all genes in child GO terms of these selected terms?)

D. Fourth, in the same analysis and hidden in methods, top-15% most variable patient genes from TCGA were intersected with the above genes from mouse (after homology mapping). The result is a small list of 94 genes that almost appears to result from data tinkering rather than principled analysis. Why top-15% and not some other number?

In particular, once the authors find mouse genes different between embryonic and post-natal timepoints (eg limma/voom $FDR < 0.05$), it would be natural to use all these genes to perform hierarchical clustering and significance tests between phenotypic groups (MITF, keratin, immune). *This whole analysis has since been removed from the manuscript. We found that after all our new computation analyses and new data, this analysis seemed extraneous from the core story of the manuscript and would be better suited supplementing other future studies in the lab.*

3. The survival analysis does not appear to consider important clinical covariates, such as tumor stage, patient age, patient sex, tumor subtypes (<https://www.ncbi.nlm.nih.gov/pmc/articles/PMC4580370/>). This information is available in TCGA. Does the Cox survival model of their gene signature outperform another control Cox model where clinical variables are listed as predictors? *We apologize for not expressing ourselves clearly in this regard. To take into account tumor stage effect on survival analysis, we performed Kaplan-Meier analysis by stratifying patients into advanced stage (III/IV) and early stage (I/II) groups. We found that advanced stage metastatic melanoma patients with upregulation of our melanoblast gene signature had worse clinical prognoses. However, this was not the case with early stage primary tumor patients (Page 5, Fig. 1 e-f). We also uncovered an upregulation of melanoblast gene expression in correlation with melanoma stage progression (Page 5, Supplementary Fig. 3. a-c). Finally, our intention of creating this dataset and subsequent analyses was not to find clinical biomarkers that necessarily outperform other known clinical covariates, but rather to explore new way of understanding the metastatic process in melanoma based on the embryonic lineage of melanocytes with a view to uncovering novel metastatic biology, mechanisms, and potential model platforms. We therefore respectively suggest that an analysis of how our signatures perform against other Cox models is irrelevant to this study and in fact would be distracting to the main ethos of the paper.*

4. Several uses of incorrect gene correlation filter are apparent [Spearman's rank of > 0.3 and p-value < 0.0001]. Multiple testing correction (FDR) should be used at all times. E.g. top of page 9, page 13 in the manuscript. *We thank the reviewer for this observation; this has now been modified throughout the entire manuscript (Fig. 7a-b, Supplementary Fig. 2, 7a-b, Supplementary Tables 2-4).*

5. Methods on selecting 149 genes (review point 1; found on page 20). "Genes differentially expressed during development were calculated by ≥ 5 -fold decrease of expression in P7 versus E15.5 (FPKM)". This five-fold increase is problematic because we do not know variation between replicates. The authors need to use a statistical model such as limma/voom or edgeR to determine significant changes between groups, fold-change alone is not sufficient. The method TREAT by the developers of Limma actually allows to combine fold-change filters with p-value filters if necessary for this analysis. *We appreciate the reviewer's comment. In response we have leveraged the generalized linear model implemented in the R package DESeq2 to select melanoblast-specific genes, selecting only significantly altered genes and using the q-value as a filter threshold.*

6. Analysis of TCGA data also seems to use non-adjusted P-values for filtering genes, FDR-adjusted values should be used instead (found on page 23). *This has now been removed from the manuscript.*

7. Why did the authors use different RNA-seq processing methods (eg Limma/voom for TCGA; DESeq2 for NCI cell lines; apparently no model for mouse transcriptomics)? Similarly, different pathway analysis methods were used (ingenuity for mouse transcriptomics, Fisher's test with MSigDB for NCI cell lines). It would look much more reliable if a consistent series of steps were applied with standard statistical filtering criteria (e.g., $FDR < 0.05$). *We thank Reviewer 3 for this suggestion and have since modified analyses for consistency. Generalized linear model in DESeq2 was used for mouse RNAseq data and a comparable analysis, linear regression model (equivalent to t-test when the independent variable is a binary factor), was used for microarray data (Pages 4-6). Analysis of KDELR3 and KDELR1 gene expression in TCGA patient data was still performed using the Bioconductor edgeR and limma R packages, as these analyses had been done by a separate collaborator using their standard analysis pipeline (Fig. 6a and 7e). Processing was consistent between KDELR3 and KDELR1 analyses to ensure that the comparison of data patterns was legitimate. Finally, to ensure consistency throughout the manuscript, all pathway analyses were undertaken using Gene Set Enrichment Analysis (GSEA), using the FDR as a threshold cut-off (Pages 8, 11-13, 19, Fig. 7a-b, Supplementary Fig. 7a).*

Again, we are indebted to the reviewers for their insightful comments and suggestions. We look forward to your decision to have our manuscript re-reviewed by the same reviewers to get their feedback on publication in *Nature Communications*.

REVIEWERS' COMMENTS:

Reviewer #1 (Remarks to the Author):

I did not have any significant comments in my previous review and my impression is the the authors have addressed the comments of the other two reviewers to the best of their ability.

Reviewer #2 (Remarks to the Author):

The authors have adequately satisfied my requests.

Reviewer #3 (Remarks to the Author):

The authors have addressed my concerns and improved their manuscript. I have no further comments at this stage.